# Diagnostic Tools for Rapid Screening and Detection of SARS-CoV-2 Infection

**DOI:** 10.3390/vaccines10081200

**Published:** 2022-07-28

**Authors:** Satish Kumar Pandey, Girish C. Mohanta, Vinod Kumar, Kuldeep Gupta

**Affiliations:** 1Department of Biotechnology, School of Life Sciences, Mizoram University (Central University), Aizawl 796004, India; 2Materials Science and Sensor Applications, CSIR-Central Scientific Instruments Organisation (CSIR-CSIO), Chandigarh 160030, India; gmohanta@csio.res.in; 3Department of Dermatology, Venerology and Leprology, Post Graduate Institute of Medical Education & Research, Chandigarh 160012, India; kumar.vinod@pgimer.edu.in; 4Russel H. Morgan, Department of Radiology and Radiological Sciences, School of Medicine, Johns Hopkins University, Baltimore, MD 21287, USA

**Keywords:** COVID-19/SARS-CoV-2, biosensors, aptamers, surface plasmon resonance, nucleic acid hybridization, immunodiagnostics techniques

## Abstract

The novel coronavirus disease 2019 (COVID-19), caused by severe acute respiratory syndrome coronavirus 2 (SARS-CoV-2), has severely impacted human health and the health management system globally. The ongoing pandemic has required the development of more effective diagnostic strategies for restricting deadly disease. For appropriate disease management, accurate and rapid screening and isolation of the affected population is an efficient means of containment and the decimation of the disease. Therefore, considerable efforts are being directed toward the development of rapid and robust diagnostic techniques for respiratory infections, including SARS-CoV-2. In this article, we have summarized the origin, transmission, and various diagnostic techniques utilized for the detection of the SARS-CoV-2 virus. These higher-end techniques can also detect the virus copy number in asymptomatic samples. Furthermore, emerging rapid, cost-effective, and point-of-care diagnostic devices capable of large-scale population screening for COVID-19 are discussed. Finally, some breakthrough developments based on spectroscopic diagnosis that could revolutionize the field of rapid diagnosis are discussed.

## 1. Introduction

After the emergence of novel coronavirus disease 2019 (COVID-19), more than a billion people have been infected with this virus and over 5.4 million people have lost their lives. The causative viral factor of this ongoing pandemic has been named SARS-CoV-2 and several other variants. SARS-CoV-2 is an RNA virus belonging to the coronavirus family of viruses (viruses with spike-like projections on the surface that appear as a crown under an electron microscope). Several species of coronaviruses are normally found in human blood circulation (HKU1, NL63, 229E, and OC43), where these viruses cause mild respiratory illness [1]. However, there have been two events wherein beta coronaviruses of animal origin jumped to humans and caused severe disease outbreaks [2]. The first one was reported in Guangdong Province, China, in 2002–2003, wherein a coronavirus of bat origin jumped to humans via palm civet cats and caused severe respiratory illness. The virus was designated as severe acute respiratory syndrome (SARS) coronavirus and infected around 8422 people, claiming 916 lives [3]. The second outbreak of coronavirus-related illness emerged in Saudi Arabia in 2012, which claimed the lives of 858 people and infected nearly 2500 people [4]. Like the previous outbreak, the virus was of bat origin and jumped to humans via dromedary camels. Incidentally, the present outbreak of COVID-19 is also caused by a bat-origin coronavirus and shares >70% homology with the SARS virus of the 2002–2003 outbreak, and hence has been named SARS-CoV-2 [5]. This review article aims to provide some recent developments in diagnostic approaches toward alleviating COVID-19 disease.

### 1.1. Onset of Disease and Transmission

The spread of the COVID-19 outbreak first indicated animal-to-human transmission observed in Wuhan City’s wet market (Huanan Seafood Wholesale Market). However, later molecular studies have concluded that the marketplace acted as a hotspot for transmission, but was not the origin of the disease [6]. The exponential rise in the number of infected patients outside the seafood market indicated human-to-human transmission, which was later confirmed by independent studies [7]. The major route of transmission of COVID-19 is exposure to respiratory droplets released by infected patients while sneezing, coughing, or even talking [8]. However, ocular transmission has been also considered as a suspected route [9,10,11]. It was soon realized that the virus could be transmitted even before the onset of symptoms. Intriguingly, some individuals also act as super-spreaders of the disease, affecting several people in a cluster [12,13,14]. The basic reproductive number (Ro) for COVID-19, which indicates its transmissibility, has been estimated to be between 2.0 and 6.47 in various models of epidemiological studies [15,16,17,18,19].

All viruses continually mutate and evolve toward better adaptation to the host cells, either through host-cell-receptor binding with higher affinity or escaping the host’s immunity, or both [20]. Being an RNA virus, SARS-CoV-2 exhibits high mutation rates, which help it to evolve and adapt at a faster rate [21]. Consequently, as of April 2022, several different variants of SARS-CoV-2 have emerged among the global population [22]. The emergence of these variants has changed the dynamics of the progression of the pandemic due to their increased transmissibility and/or effective host immune escape. Although some success has been achieved through large-scale global vaccination drives against SARS-CoV-2, the continual diversification of the virus poses a formidable challenge [23,24]. The emergence of a more transmissible and/or virulent strain of SARS-CoV-2 could then fuel the pandemic further.

The World Health Organization (WHO) has been tasked with the continuous monitoring, reporting, and assessment of the virulence and transmissibility of the newly emerged SARS-CoV-2 variant in populations around the globe. So far, several SARS-CoV-2 variants have been reported, most of which possess similar transmissibility and virulence as those of the original strain. However, three variants of SARS-CoV-2, namely delta (B.1.617.2) and omicron (BA1 and BA2), have been designated as “variants of concern (VOCs)” by the WHO, and each has been attributed with causing fresh waves of infections in their originating countries (Source: https://www.who.int/activities/tracking-SARS-CoV-2-variants (accessed on 20 April 2022)). While the delta variant proved to be highly virulent and caused severe disease and death, the omicron variant is considered to be ~70 times more transmissible than the original SARS-CoV-2 strain [25,26]. Thus, the fate of the pandemic relies on the rate of public healthcare intervention, such as early diagnosis, isolation of the infected population, and the rate of virus diversification.

### 1.2. Clinical Manifestation

Similar to previous outbreaks of SARS viruses, COVID-19 also attacks the respiratory system and thus presents symptoms of respiratory illness [27]. The clinical manifestation of COVID-19 ranges from typical, flu-like mild symptoms, such as fever, coughing, and fatigue, to more severe symptoms, such as pneumonia, multiple-organ failure, and even death. Other symptoms that are less frequently presented with COVID-19 include gastrointestinal discomfort, sputum production, headache, hemoptysis, diarrhea, dyspnea, and lymphopenia. Fortunately, more than 90% of affected individuals suffer from only mild symptoms that resolve within a week by themselves, and the patient makes a total recovery within 2–3 weeks. However, in some patients, the disease progresses to severe conditions requiring oxygen supplementation and mechanical ventilators [28]. It has been identified that an individual’s immunity plays a very critical role in the progression of COVID-19 disease to the severe form. Elderly people (>60 years), people with additional comorbidities, such as cancer, hypertension, cardiovascular disease, respiratory diseases, liver diseases, obesity, diabetes, and renal diseases, and immune-compromised people are particularly vulnerable to developing severe symptoms [29]. Smoking has also been included as a potential risk factor [30,31]. In contrast to adults, the clinical manifestation of COVID-19 in pediatric patients and adolescents is relatively mild, with very few cases requiring hospitalization and intensive care. The symptoms at onset are similar to those of adults, including fever and coughing that rapidly resolve. However, unlike adults, several COVID-19-positive children have developed symptoms of pediatric multisystem inflammatory syndrome (PMIS) [32]. Intriguingly, gender impact studies on COVID-19 have indicated that males are relatively more vulnerable than females to developing the severe form [33,34]. This difference in COVID-19 progression has been attributed to the sex hormone, testosterone. Decreased levels of testosterone in males have been found to be linked to the disease severity of COVID-19 [35,36].

## 2. Diagnosis

The novel SARS-CoV-2 virus is highly contagious and virulent, as indicated by the severity and rapidity of its transmission. The prolonged incubation period in affected individuals, combined with the virus’ ability to remain viable for long periods of time on different surfaces, further intensify the contagion. One of the major challenges in containing the spread of COVID-19 is the transmission of the virus by pre-symptomatic and asymptomatic carriers. Several case studies have revealed that the virus could be transmitted by an infected individual even before the onset of symptoms [37]. Therefore, timely and accurate diagnosis of COVID-19 in the suspected population is very crucial to contain the silent spread of the disease [38]. Additionally, timely diagnosis is very critical for monitoring disease progression and management in high-risk patients. Furthermore, timely and mass diagnosis of the disease in the general population would facilitate epidemiologists and healthcare agencies in estimating the disease burden and its progression [39]. This will help to frame the necessary regulatory steps required to contain the further transmission of the disease, such as zone-wise lockdowns, the creation of buffer and containment zones, travel restrictions, etc. Consequently, several industries and academia are developing rapid, convenient, and point-of-care diagnostic kits for the accurate detection of the SARS-CoV-2 virus [40,41,42]. These approaches to COVID-19 diagnosis could be divided into three major categories: nucleic acid-based diagnosis, serology-based diagnosis, and imaging-based diagnosis.

### 2.1. Nucleic Acid-Based Diagnosis

The detection of the genomic regions of pathogens (bacteria, viruses, parasites, etc.) is the basis for diagnosis in all nucleic acid-based detection assays. The detection could be achieved either by amplifying a specific region of the genome through the polymerase chain reaction (PCR) or by hybridizing a complementary oligonucleotide reporter probe with a specific region of the pathogen DNA (or RNA) [43]. A third, less commonly adopted approach is to culture the pathogen and carry out whole-genome sequencing using high-throughput sequencing instruments. However, sophisticated instrumentation and its expensive nature limit the application of such high-throughput sequencing methods in clinical diagnosis [44]. It is largely limited to research facilities, where it is used to understand the basic molecular biology of pathogens.

#### 2.1.1. Reverse-Transcription–Polymerase Chain Reaction (RT-PCR)

The RT-PCR is the most widely utilized diagnostic tool for the detection of retroviral infection [45]. It is a two-step process, first consisting of converting the viral RNA into complementary DNA (cDNA) strands and then amplifying a tiny portion of the viral cDNA using a primer pair. Both processes can be run independently and simultaneously to reduce the detection time, cross-contamination, and handling errors. Additionally, using two or more primer pairs to amplify different regions of the genome significantly increases the specificity of the method. RT-PCR is considered as the gold-standard test for COVID-19 diagnosis, as it can also evaluate the viral load of the infection and therefore assist in disease prognosis [46]. One of the earliest RT-PCR kits for COVID-19 diagnosis was commercialized by LabCorp (The EUA certificate is available at https://www.fda.gov/media/136151/download (accessed on 27 December 2021)), after which several manufacturers followed suit. Despite being widely used and reportedly having excellent specificity, RT-PCR still suffers from a few drawbacks, such as sub-optimal sensitivity, the requirement of skilled technicians, expensive consumables, and being time-consuming. In the case of mass screening, a day or two is required to obtain the diagnosis report. The sensitivity of RT-PCR depends on a number of factors, such as the sample type, original viral load, as well as the reporting technique. In a case study of around 4880 people who had reported suspected COVID-19 symptoms, it was found that the positivity rate was 40% for nasopharyngeal swabs, 50% for sputum sample, and 80–100% for bronchoalveolar lavage fluids. Despite its low sensitivity, the preferred method of sampling is the collection of viral particles from the upper respiratory tracts through nasopharyngeal swabs due to higher patience compliance and lower risk of exposure of the health worker to viral aerosols. Recently, COVID-19 was successfully diagnosed through RT-PCR using self-collected saliva samples with high sensitivity (97%). This indicates that saliva could be utilized as a promising sampling alternative to the presently utilized nasopharyngeal swabs for the diagnosis of SARS-CoV-2 infection [47,48].

The fact that SARS-CoV-2 viral particles are predominantly transmitted by exhaled breath and droplets has motivated researchers to explore exhaled breath condensate (EBC) as an alternate method of sampling to nasopharyngeal swabs. EBC has greater patient compliance and provides minimal risk of exposure to the healthcare provider. Studies related to the detection of SARS-CoV-2 using EBC as a sampling method have revealed a good correlation between the viral load in the breath and the disease’s progression. Interestingly, EBC sampling will also be able to identify super spreader events by analyzing the viral load of the individual in exhaled breath [49]. Although the sensitivity of EBC is lower than that of nasopharyngeal swabs, it provides excellent patient compliance, particularly for elderly patients and patients on ventilators [49,50,51].

Despite its functional superiority over other diagnostic assays, RT-PCR suffers from a few shortcomings, such as higher incidences of false-negative results, higher susceptibility to inhibitors, and also being affected by sampling and storage conditions [52,53,54]. This could increase the risk of incorrect diagnosis and further aggravate the current pandemic situation [55]. As a result, it has been recommended to combine RT-PCR with other diagnostic assays, such as chest CT scanning, for the final diagnosis of a suspected case [56,57].

To overcome the shortcomings of RT-PCR, digital droplet PCR (ddPCR) has been suggested as an alternative diagnostic tool for COVID-19 diagnosis [58]. In ddPCR, a PCR reaction mixture containing target-specific primers (against orf1ab or the nucleocapsid gene of SARS-CoV-2) and DNA-binding fluorescent dyes or Taqman probes are partitioned into water-in-oil emulsion droplets. As compared with RT-PCR, ddPCR offers several advantages including lower sensitivity to inhibitors, absolute quantification of target DNA, higher sensitivity and precision, and better stability [59]. In fact, several studies have demonstrated the superior diagnostic capability of ddPCR over conventional RT-PCR [60].

#### 2.1.2. Isothermal Nucleic Acid Amplification

One of the inherent challenges of RT-PCR-based assays is the requirement for precise temperature control and thermal cycling steps at different values. This variation and precise control of temperature necessitate sophisticated instrumentation and longer duration (>3 h.) for analysis in RT-PCR. Consequently, several isothermal nucleic acid amplification methods have been reported wherein the amplification occurs at a fixed temperature and therefore significantly reducing the analysis time [61]. Several such isothermal techniques are being developed for COVID-19 diagnosis, with the major focus now toward developing point-of-care (POC) devices [62].

##### Reverse Transcription Loop-Mediated Isothermal Amplification (RT-LAMP)

Unlike RT-PCR, loop-mediated isothermal amplification (LAMP) utilizes four to six primers for amplifying a specific region of the target DNA at a constant temperature. The use of additional primers provides exceptional specificity, while amplification at a single temperature allows rapid analysis (typically within 30 min). The amplified products could be detected visually using chromogenic reagents or through turbidity production during the accumulation of magnesium pyrophosphate precipitates as the byproduct of the reaction. The real-time monitoring of the amplification is carried out by DNA-intercalating fluorescent dyes. Since LAMP amplification takes place at a fixed temperature and can be traced visually (turbidity), it is also suitable for the development of point-of-care (POC) molecular diagnostics [63]. In this direction, Abott Diagnostics have commercialized a rapid POC device based on RT-LAMP for the diagnosis of SARS-CoV-2 RNA in the upper respiratory swabs (the EUA certificate is available at https://www.fda.gov/media/136522/download (accessed on 27 December 2021)) [64]. The diagnosis is very rapid and could be completed within 13 min. However, only one sample per run limits the scalability of the technique. Traditionally, LAMP was carried out using Bst DNA polymerase, which is a DNA-dependent DNA polymerase. This meant that a preliminary step of converting viral genomic RNA into cDNA would be necessitated. However, using WarmStart RTx Reverse Transcriptase from New England Biolabs, Inc., Massachusetts, US, cDNA synthesis and subsequent LAMP amplification could be carried out in a single tube [65]. WarmStart RTx Reverse Transcriptase is an in silico-designed unique RNA-directed DNA polymerase with reversible reverse transcriptase activity, wherein a coupled aptamer inhibits the reverse transcriptase activity below 40 °C, thus allowing the room-temperature set-up of the reaction.

In a major advancement to LAMP-based diagnostic assays, recently, the recombinase polymerase assay (RPA) was combined with LAMP in a two-stage, closed-tube approach with POC capability [66]. The combined approach is called RAMP and utilizes additional recombinase polymerase amplification along with LAMP. Combining the two approaches afforded exceptional specificity and sensitivity to COVID-19 diagnosis. Particularly under low-viral-load conditions, the combined approach was found to be 100 times more sensitive than LAMP alone. Several studies have reported better or comparable sensitivity and specificity of LAMP as compared with RT-PCR for the diagnosis of COVID-19. Together with its simplicity, rapidity, and POC capabilities, RT-LAMP is an excellent alternative to traditional RT-PCR.

In yet another technological advancement, researchers were able to successfully integrate the RT-LAMP assay on a microfluidic chip platform [67,68]. Such a development not only allowed point-of-care diagnosis, but also minimized the cost of operation as only a small volume of PCR reaction mixture (~5 µL) would be required. Furthermore, chip-based analysis also makes the RT-LAMP assay suitable for mass screening, which, in a conventional setup, is restricted to laboratory procedures.

##### Transcription-Mediated Amplification (TMA)

Transcription-mediated amplification is an isothermal amplification technique for RNA targets and utilizes a retroviral reverse transcriptase and T7 RNA polymerase enzyme to efficiently amplify a particular target region of pathogenic RNA. The efficiency of TMA is reported to be higher than that of RT-PCR and it has been successfully employed for multiple-pathogen detection [69]. The principle of amplification is depicted in Figure 1. Using the principle of TMA, Hologic Inc. developed the “Hologic Aptima SARS-CoV-2 Assay”, which could detect the SARS-CoV-2 virus in both the RT-PCR and TMA format (the EUA certificate is available at https://www.fda.gov/media/138096/download (accessed on 27 December 2021)). A comparative study between the two reported the superior sensitivity of TMA for COVID-19 diagnosis as compared with RT-PCR. One of the reasons for having such a high sensitivity was the generation of thousands of transcribed copies of the target, which further acted as transcriptional templates for generating more copies. Another noted advantage of TMA over RT-PCR was the ease of operation. The collected samples could be easily loaded into the Panther device for TMA without requiring any prior extraction, as in the case of RT-PCR, thus allowing high-throughput screening, minimal exposure risk, and efficient workflow [70].

In a recent study, it was reported that Hologic’s Aptima SARS-CoV-2 assay performed even better in comparison with two other commercial RT-PCR assays, particularly in dealing with pooled samples (10 to 1). Furthermore, the turnaround time for sample analysis using Aptima was significantly lower than that for RT-PCR, making it suitable for screening large amounts of sample [71]. To date, several other commercial firms have adopted and marketed the chemiluminescence-based TMA assay as a diagnostic tool for the detection of the SARS-CoV-2 virus (the EUA certificate for various manufacturers is available at https://www.fda.gov/medical-devices/coronavirus-disease-2019-covid-19-emergency-use-authorizations-medical-devices/in-vitro-diagnostics-euas-molecular-diagnostic-tests-sars-cov-2 (accessed on 5 January 2022)).

##### Rolling Circle Amplification

Rolling circle amplification (RCA) is another isothermal amplification technique that has garnered significant research interest due to its high sensitivity, robustness, and specificity as compared with traditional RT-PCR [72]. So far, the RCA-based detection kit for COVID-19 has not been approved yet, but some developments have been reported in this direction. Hsu and co-workers reported a RCA-based diagnostic assay for the identification of three respiratory viruses: SARS-CoV-2, Influenza A (H1N1pdm09), and Influenza B (Victoria Lineage), on a single platform. The platform could sensitively detect picomolar concentrations of synthetic viral DNA within 30 min with very high specificity. Furthermore, the reaction could be visually monitored using a pH-sensitive dye, such as phenol red in this case [73]. Recently, the RCA platform was developed in a lateral flow assay (LFA) format for the detection of the SARS-CoV-2 virus [74]. Apart from the benefits of the LFA format, such as point-of-care detection, cost-effectiveness, reliable results, and robustness, the detection platform also exhibited highly sensitive and specific detection capacity for target DNA, with a limit of detection (LOD) of about 1.1 pM (or 1.3 × 10^6^ copies per reaction).

##### CRISPR-Based Assays

CRISPRs are “Clustered Regularly Interspaced Short Palindromic Repeats”. As the name suggests, CRISPRs are the specialized sequences of DNA found in prokaryotic organisms, such as bacteria and archaea. Bacteria use CRISPR-derived RNA to chop off and destroy foreign DNA, such as that of bacteriophages. They do so by recruiting the family of CRISPR-associated enzymes, such as Cas9. Therefore, CRISPR is also known as the CRISPR–Cas9 system. These enzymes act as precision molecular scissors and can induce sequence-specific double-strand nicks in foreign DNA [75].

Recently, the United States Food and Drug Administration (US FDA) provided approval to the world’s first CRISPR-based diagnostics assay for SARS-CoV-2 detection to Sherlock biosciences (the EUA certificate is available at https://www.fda.gov/media/137746/download (accessed on 5 January 2022)). The detection could be performed within an hour, starting from RNA extraction, isothermal amplification using the recombinase polymerase assay (RPA), followed by the detection of amplified RNA targets using Cas13-enabled excision of reporter RNA. The assay gives a visual read-out on a paper-based dipstick technique [76]. In another approach, Mammoth Biosciences developed a CRISPR–Cas12a system for the diagnosis of COVID-19 using SARS-CoV-2 RNA in a technique called DNA Endonuclease-Targeted CRISPR Trans Reporter (DETECTR) (the EUA certificate is available at https://www.fda.gov/media/139937/download (accessed on 5 January 2022)). In this approach, the target RNA is first amplified using RT-LAMP, followed by confirmation through a Cas12a-assisted CRISPR system to detect the amplified RNA targets. Positive confirmation was attained visually [77]. In a similar approach, Sun et al. combined reverse transcription and recombinase polymerase isothermal amplification (RT-RPA) and DETECTR technology to develop an easy-to-use and one-tube format for the detection of SARS-CoV-2. Interestingly, the combined approach gave an ultrasensitive detection limit of just 2.5 copies of DNA per microliter of sample, and the entire process could be completed within 50 min. Furthermore, the detection platform also exhibited very high specificity by successfully identifying the SARS-CoV-2 virus from seven other human coronaviruses, as well as H1N1 influenza virus [78].

The simpler instrumentation, ease of operation, and capability to be carried out in formats such as paper and microfluidic channels without loss of sensitivity and specificity make CRISPR-based assays excellent for point-of-care diagnosis. Despite their approved success, the Sherlock and Mammoth CRISPR assays still depend on prior nucleic acid extraction and isothermal amplification, thus limiting their usage in laboratory setups. The next generation of CRISPR assays are, therefore, focused toward developing a truly POC platform wherein the detection could be conducted with intact patient samples, such as nasopharyngeal swabs, buccal swabs, or, at best, saliva. In this direction, Cardea Bio Inc., a San Diego-based biotechnology company, has reportedly developed a graphene field-effect transistor (gFET)-based diagnostic kit for the detection of SARS-CoV-2, which relies on the CRISPR–Cas9 system for target identification [79]. Intriguingly, the detection could be conducted with unamplified samples and, therefore, offer true POC capability. The technology is based on their previous work on the detection of a genetic disorder: Duchenne muscular dystrophy [80]. The development of another POC, electrochemical, and microfluidic CRISPR–Cas13 system for the detection of unamplified SARS-CoV-2 RNA has been reported [81]. The system utilizes the non-specific collateral cleavage capability of the Cas13 enzyme to cleave a reporter RNA. On positive identification, the reporter RNA is not excised and is, therefore, recognized by a glucose oxidase-tagged antibody that generates hydrogen peroxide for electrochemical detection. The concentration of generated hydrogen peroxide is inversely proportional to that of the target RNA. For a one-step approach for the detection of unamplified SARS-CoV-2 RNA, a primer-free CRISPR isothermal amplification assay has been reported [82]. The detection takes place at ambient temperature with attomolar (10^−18^ M) sensitivity. A positive signal is identified by the fluorescent read-out.

#### 2.1.3. Nucleic Acid Hybridization Microarray Assays

DNA microarrays are well-suited for the rapid and high-throughput detection of SARS-CoV-2 nucleic acid. The assay depends on the specific hybridization of viral DNA with immobilized complementary sequences to give a positive signal, as illustrated in Figure 2 [83,84]. The microarray techniques have been utilized previously for the identification of seven different coronaviruses causing animal and human respiratory diseases. It has been also utilized for the identification and genotyping of single-nucleotide polymorphism (SNP) mutations in coronavirus, thus highlighting its applicability for the diagnosis of SARS-CoV-2.

PathogenDx, Arizona, a US-based biotechnology company, has reported a DirectX-Rv diagnostic test kit for SARS-Cov-2 detection. The detection platform utilizes a multiplex RT-PCR and DNA microarray technique in tandem for the sensitive, specific, and quantitative detection of the SARS-CoV-2 virus. Interestingly, the detection method can also differentiate between the closely related SARS-CoV-1 and SARS-CoV-2 viruses due to multiple target gene identification [85].

One of the most significant aspects of the DNA microarray technique is its capability to simultaneously identify and distinguish several different pathogens or different strains of a pathogen on a single platform. The Combinatorial Arrayed Reactions for Multiplexed Evaluation of Nucleic acids (CARMEN)–Cas13 platform can be used for the multiplex detection of around 169 human*associated viruses [86]. Recently, detection capability for SARS-CoV-2 has been added to the array, taking the tally to 170 pathogens. The incorporation of the CRISPR–Cas13 system with the microarray format gives robustness and very high specificity to the assay. Recently, another DNA microarray-based detection technology was reported by Damin et al., called Covid Array [87]. In this technique, N1 and N2 target genes of SARS-CoV-2 are first reverse-transcribed into cDNA and amplified through RT-PCR using fluorescent-dye-labeled primers. The amplicons are then subjected to solid-state hybridization over a Si/SiO_2_ chip, and the fluorescent signal from captured amplicons is determined to identify positive or negative results. Remarkably, the microarray system had an ultrasensitive detection limit of just 1.16 copies/µL and was also able to correctly identify and assign 12 COVID-19 positive nasopharyngeal swab samples previously deemed as negative by RT-qPCR [87]. The authors also developed a novel RNA isolation protocol called RNAGEM to extract and purify SARS-CoV-2 viral RNA in under 15 min from swab samples. Recently, they marketed this protocol as an RNA/DNA isolation kit under the tradename RNAGEM V (Available at www.microgembio.com (accessed on 5 January 2022)).

#### 2.1.4. Nucleic Acid Sequencing

In any pandemic response, nucleic acid sequencing is, perhaps, the first-line molecular technique used to identify and characterize the causative pathogen. Although nucleic acid sequencing techniques are generally not preferred for clinical diagnosis due to the expensive instrumentation, expert handling, and high operational cost involved, few studies have been reported in this regard. For example, Moore et al. utilized two complementary sequencing techniques—the amplicon and metagenomic MinION sequencing techniques—to deduce the genome of the SARS-CoV-2 virus and other microbiomes present in the nasopharyngeal swabs of COVID-19 patients [88]. For amplicon-based sequencing, 16 different primers were designed that sequentially spanned the entire SARS-CoV-2 genome in roughly 1000 bp amplicons with around 200 bp overlaps so that sequences can be assembled from the amplicon data. The primers were designed from the conserved regions of the SARS-CoV-2 genome available on GSAID. The amplicons were then sequenced using MinION. For metagenomic sequencing, sequence-independent single primer amplification (SISPA), a reliable technique to sequence and identify unknown pathogens, was used. The metagenomic sequencing approach is particularly relevant in the case of COVID-19, since secondary infection by other pathogens markedly influences the disease’s progression from mild to more severe [89]. Similarly, there has been increasing evidence of a linkage between the severity of gastrointestinal symptoms and the gut microbiome in COVID-19 patients [90]. An understanding of the background microbiome or co-infection of a COVID-19 patient will facilitate clinicians to design more aggressive and effective treatment regimes, and therefore reduce morbidity and mortality. A next-generation shotgun metagenomic sequencing technique has been developed and is now commercially available from Illumina. The technique can not only identify different strains of coronavirus, but also detect multiple pathogenic microorganisms in a highly complex sample.

Recently, a rapid and inexpensive whole-genome sequencing (WGS) of SARS-CoV-2 was reported, which has cut the time required for traditional sequencing by half [91]. The authors did so by intelligently creating multiplex 1200 bp tiled amplicons in two pools and then preparing the combined sequencing library using the Oxford Nanopore Rapid Barcoding Kit. The technique holds promise for rapid and cost-effective sequencing-based clinical diagnosis.

The major emphasis of various sequencing techniques is identifying the predominant strain or variant of SARS-CoV-2 circulating among the population, rather than simply being used as a diagnostic test, per se. The continuous emergence of various variants of concern (VOCs) of SARS-CoV-2 around the world has re-emphasized the need for simultaneous mass testing and genomic surveillance through nucleic acid sequencing techniques [20]. In this direction, Yermanos and co-workers developed a high-throughput platform, DeepSARS, which is capable of the simultaneous diagnosis and genomic surveillance of the SARS-CoV-2 virus. The platform utilizes integrated molecular barcoding, deep sequencing, and a computational phylogenetic approach to successfully detect and identify the variant of SARS-CoV-2 [92]. Single-nucleotide mutations (also called single-nucleotide polymorphisms (SNPs)) and deletion mutations are commonly found in the gene responsible for spike proteins of various SARS-CoV-2 variants. Two such mutations are N501Y, found in Alpha, Beta, and Gamma, and E484K, found in the Beta and Gamma VOCs of SARS-CoV-2. In addition to N501Y, the Alpha variant also exhibits a deletion mutation (ΔHV69/70). Utilizing this information, Schindler and co-workers developed a SNP-specific portable RT-qPCR, *peak*PCR, to rapidly detect and identify SARS-CoV-2 VOCs [93]. Remarkably, the device can accomplish 20 reactions in less than 40 min, while reaction mixtures could be transported as pre-loaded cartridges under ambient conditions, thus extending its applicability in resource-constrained regions. Further advancing on this approach, Allen et al. reported a pyrosequencing protocol that can simultaneously detect and classify SAR-CoV-2 into various VOCs by monitoring more than 65 spike gene mutations [94]. The protocol can identify all of the major VOCs currently circulating among the US population.

### 2.2. Serological and Immunological Assays

A serological-based diagnostic assay utilizes patient blood samples (plasma or serum) to identify viral proteins or antibodies. Sometimes, other sample types, such as sputum, saliva, or other bodily fluids, are also broadly classified into serological tests as the method of diagnosis remains the same. In response to SARS-CoV-2 infection, the humoral immune system of the body produces neutralizing antibodies against the viral proteins. In serological tests, these antibodies are captured and quantitatively detected for diagnosis. Interestingly, the type of antibody detected in serological tests also reveals the timeline of the infection and, therefore, provides valuable information regarding the epidemiology of the disease and immunity status of the population [95]. For example, IgM antibodies are the first line of active defense against the invading pathogen, and their presence indicates recent infection, while IgG antibodies are produced late in the response and are generally involved in clearing the pathogen. IgG antibodies are also produced in case of a second infection by the same pathogen. Thus, the presence of IgG indicates long-term immunity in the population [96,97].

Alternatively, serological tests can also detect SARS-CoV-2-specific proteins (also called antigens) for diagnostic purposes. There are around 29 proteins of the SARS-CoV-2 virus, of which 4 are structural proteins (E, M, N, and S), and the rest are non-structural proteins. The structural proteins N and S, which are highly expressed during infections, are ideal targets for serological testing. However, the mere presence of these proteins alone would not constitute an active infection and, therefore, should be corroborated clinically. The receptor-binding domain (RBD) of the S protein, which is crucial in viral attachment, is also another target of interest for serological testing [98]. As the RBD domain is usually not conserved across different species of human coronaviruses, it serves as a suitable target for the highly specific detection of the SARS-CoV-2 virus [99]. Usually, neutralizing antibodies used in therapies often target the RBD domain of the spike protein to disrupt its interaction with the host receptor; therefore, an estimation of anti-RBD antibodies will also reveal the status of the immune protection of the individual [100]. Consequently, several serological studies have been reported where antibodies against the RBD domain have been determined, particularly IgG antibodies, which may confer short-term immunity to the individual [101].

In general, serological tests are rapid, cost-effective, and have on-site detection capability; therefore, they are ideally suited for high-throughput testing in pandemics [98]. However, serological tests suffer from low sensitivity and specificity as compared with their molecular diagnostic counterparts. Therefore, many countries have adopted the use of RT-PCR as a confirmatory test in cases where serological tests yield negative results, despite patients exhibiting suspected symptoms of COVID-19. There are three potential pitfalls for serological tests wherein they may produce false-negative or false-positive results: (1) a lag in antibody production in some individuals post-infection may give false negative results, despite being positively identified through molecular diagnostic techniques; (2) patients may produce false-positive serological tests, even after the clearance of the pathogen after a mild infection, while this may produce a negative result in a molecular assay; and (3) the low sensitivity and specificity of the serological test may produce cross-reactivity and therefore yield a false-positive diagnosis. Nonetheless, the rapid screening capability, cost-effectiveness, and information regarding the immune status of a population make serological tests an indispensable tool for monitoring the epidemiology of pandemic diseases [102]. Several methods for serological testing are available on the market. They include the traditional enzyme-linked immunosorbent assay (ELISA), immunochromatographic lateral flow assay (LFA), and biosensors.

#### 2.2.1. Enzyme-Linked Immunosorbent Assay (ELISA)

The ELISA is perhaps the most widely adopted technique for the serological testing of various diseases. It is a microplate-based testing platform for the identification of disease antigens or antibodies in the patient sample (serum, plasma, saliva, etc.). It typically requires 1–5 h and produces a result qualitatively or quantitatively [103].

For the detection of SARS-CoV-2 infection, viral proteins, typically S or N proteins, are coated on the base of the microplate wells. The antiviral antibodies, if present in the sample (serum) of the suspected patient, are allowed to form a protein–antibody complex in the well. After proper washing, the bound protein–antibody is detected and quantified using another reporter–antibody complex, giving a colorimetric read-out. Multiple samples can be analyzed by ELISA in one go. Furthermore, the identification of several antigenic proteins or different pathogens (i.e., multiplexing) is also possible in ELISA. The automation of the method could produce high-throughput analysis and POC detection capability in ELISA [104]. One of the challenges in using viral proteins for ELISA is the lack of antibodies for analysis. An effective solution for this is to develop synthetic affinity ligands that could selectively bind to viral proteins. In one such approach, two DNA aptamers were developed by the Systemic Evolution of Ligands by Exponential Enrichment (SELEX) method against the receptor-binding domain (RBD) of the S1 viral protein. The aptamers are 51 and 67 nucleotides long and have high affinity for the RBD domain [105]. In a similar study, a 23-mer blocking peptide against the RBD of the SARS-CoV-2 S protein was developed. The peptide was derived from the peptidase domain (PD) α1 helix of the angiotensin-converting enzyme 2 (ACE2) receptor and could bind the RBD of the S1 protein with high affinity [106]. The aptamers and peptides are ideally suited as probes for serological testing due to their small size, easy synthesis, and functional modifications.

In an interesting approach, the magnetic-bead-based ELISA assay was reported by Huergo and co-workers [107]. In this approach, the entirety of the ELISA assay, including the binding of the analyte antibody, binding with the enzyme-linked secondary antibody, and chromogenic substrate development, takes place on the surface of magnetic beads that are suspended in the liquid. Because of the close proximity and homogenous mixing of the analyte antibody and antigen on magnetic beads (both suspended in liquid), the whole ELISA process could be completed remarkably within 15 min. Additionally, the sensor setup requires a small amount of sample and consumables, and has a quick turn-around time.

#### 2.2.2. Lateral Flow Assay (LFA)

The immunochromatographic lateral flow assay (LFA), developed by Cellex Inc., was the first serological test approved by the United States Food and Drug Administration (US FDA) under the emergency usage authorization (EUA) (the EUA certificate is available at https://www.fda.gov/media/136622/download (accessed on 9 February 2022)). It is one of the rapid diagnostic techniques for qualitative serological analysis [108]. The time required for obtaining the result is around 10–30 min, and the test could be performed in a POC setup. In LFA, the viral antigens are immobilized as a band over a PVDF membrane, and the patient sample is drawn over it through capillary action. If the patient sample contains any antiviral antibodies, it will form a complex at the immobilized antigen band, which is then colorimetrically detected via a secondary reporter antibody. The working principle of LFA is depicted in Figure 3. The rapid analysis, inexpensiveness, and no requirement for trained personnel make the LFA suitable for the large sample screening required in the case of a pandemic [109].

Recently, Srivastav and co-workers combined the robust platform of LFA with the ultrasensitive SERS technique for the detection of SARS-CoV-2-specific IgG/IgM in patient serum [110]. In this approach, instead of using conventional colored beads for visual indication, gold nanostar SERS tags are utilized. The SERS tag-based detection resulted in an over-seven-fold increase in the overall sensitivity of the LFA sensor device as compared with the conventional LFA. Consequently, a very low LOD of 100 fg/mL is achieved, as compared with 1 µg/mL for the conventional LFA.

#### 2.2.3. Luminescent Immunoassay

In luminescent immunoassays, the basic principle of immunoassay remains the same as that of ELISA. However, fluorescent or chemiluminescent tracer probes are used for optical read-out, instead of traditional chromogenic reagents. The result is enhanced detection capability and a very low LOD. One of the major advantages of using luminescent signals is the massive improvement in the signal-to-noise ratio, thus allowing picomolar, or even femtomolar, detection capability [111]. Ortho-Clinical Diagnostics, Inc. USA, developed a chemiluminescent immunoassay (CLIA) to test the presence of IgG antibodies in patient serum. The method is designed to work on the VITROS Diagnostics platform and is reported to have 87.5% sensitivity and 100% specificity. The time taken to generate the first result is approximately 48 min (the EUA certificate is available at https://www.fda.gov/media/137361/download (accessed on 9 February 2022)). In an interesting approach, Roche Diagnostics, US, devised an electro-chemiluminescence (ECLIA) that decreased the CLIA assay time by half to less than 20 min. The assay identifies IgM and IgG antibodies in the patient serum with very high sensitivity and specificity (the EUA is available at https://www.fda.gov/media/144037/download (accessed on 9 February 2022)). Another portable microfluidic immunoassay was reported by Lin et al., which could detect serum IgM and IgG antibodies against SARS-CoV-2 within 15 min [112].

#### 2.2.4. Protein Microarray

Similar to DNA microarrays, protein microarrays could also be used in the detection of SARS-CoV-2 viral proteins or antibodies in patient samples. For example, to identify the antigenic determinants in COVID-19 patients and their antibody profiles, Jiang et al. created a proteome microarray using 18 out of 28 predicted proteins [113]. It was found that almost all of the patients had IgM and IgG antibodies against N and S1 viral proteins of SARS-CoV-2, thus identifying them as suitable antigens for the development of immunoassays. However, due to the availability of less expensive rapid immunoassay techniques, these are seldom used for diagnostic purposes.

#### 2.2.5. Agglutination Assay

Agglutination assays are routinely performed in hospital laboratories for blood typing and the identification of antibodies in patient serum or plasma. In the case of column-based agglutination assays, reagent red blood cells (RRBCs) are mixed with antibody-containing serum/plasma on the top of a filtration gel card and then allowed to interact for 15–20 min. After 20 min, the gel is spun in a centrifuge to allow the mixture to diffuse into the gel. In the case of positive recognition, agglutinated RBCs do not enter the gel and exhibit a sharp band at the top. Such agglutination assays are very simple, rapid (10–30 min), cost-effective, and true point-of-care methods, and they have been extensively used for screening HIV-positive patients [114]. A similar detection system was created by Alves et al., wherein RBCs were functionalized with a peptide–antibody conjugate to identify the SARS-CoV-2 spike protein antibody in patient sera [115]. The peptide was produced from a region of the SARS-CoV-2 spike protein, while the antibody was against the surface glycoprotein of RBCs.

#### 2.2.6. Neutralization Assay

In the neutralization assay, the ability of patient antibodies to prevent viral infection in cultured cells is assessed. The TMPRSS2-expressing VeroE6 cells have been found to be highly susceptible to SARS-CoV-2 infection, therefore making them suitable for neutralization assays [116]. The assay elucidates the immune status of an individual by assessing whether the antibodies are still effective against the virus, even after the infection has been cleared. The assay requires a BSL 3 or higher laboratory setup. The test is performed using a patient sample (whole blood, serum, or plasma), diluted to different extents, and added to cultured cells. If neutralizing antibodies are present in the sample, the threshold level at which they are able to block viral replication is measured. The whole process could take 3–5 days, but could be completed within hours by combining imaging or fluorescence techniques [117,118]. Although neutralization assays are not routinely used for diagnostic purposes, they are essential in determining therapeutic interventions, such as convalescent plasma therapy and vaccine development.

### 2.3. Imaging-Based Diagnosis

#### 2.3.1. Chest Computed Tomography (CT) Scan

One of the hallmark clinical symptoms of COVID-19 is the development of pneumonia, particularly in high-risk and elderly patients. Chest radiography (CT scan) is, therefore, used as one of the primary techniques to identify abnormalities in suspected patients. Typical radiological abnormalities of SARS-CoV-2 infections include bilateral, multiple lung lesions with peripheral distribution, ground glass opacities, lobular, and segmental areas of consolidation. Pathological studies of deceased patients have revealed bilateral diffuse alveolar damage (DAD) with cellular fibro-myxoid exudates, indicating acute respiratory distress syndrome. The pathological features of COVID-19 are starkly similar to those of SARS and MERS (Middle East Respiratory Syndrome). Consequently, chest CT scans are also considered as an auxiliary diagnostic technique. Intriguingly, several comparative studies between CT scans and RT-PCR for COVID-19 diagnosis have indicated the higher sensitivity of CT scans in identifying positive cases, particularly in the initial period of infection. Combining computational aid with CT scans, such as image processing and machine learning techniques, could further improve the sensitivity of detection and automation [119]. For instance, using a convolutional neural network (CNN), Sarki and co-workers developed an automated COVID-19 detection algorithm based on chest X-ray images. The algorithm could distinguish between normal and COVID-19 chest X-rays with 100% accuracy, while the distinction between normal, COVID-19, and pneumonic chest X-rays was reported with 93.75% accuracy [120]. Using a much larger data set of over 10,000 chest X-rays (CXR) and the Deep Learning Method (DLM), Mitra and co-workers achieved a remarkable accuracy of 96.43% for the positive identification of COVID-19. Interestingly, the algorithm is also able to distinguish between COVID-19 and non-COVID-19 pneumonia with 97% accuracy [121].

Therefore, considering the high false-negative rate of RT-PCR and serological tests in the initial phase, abnormal CT scan findings could be a basis for reducing misdiagnosis. However, radiological interventions for COVID-19 also overlap with the clinical features of other infections. Thus, the observations from CT scans should be corroborated with clinical diagnosis. On the other hand, chest CT scans could provide rapid identification and help clinicians make reliable decisions for treatment planning in suspected cases, even before the clinical symptoms appear.

#### 2.3.2. Lung Ultrasound (LUS)

Recently, several studies have also highlighted the potential of lung ultrasound (LUS) in the early detection of COVID-19 [122,123]. In comparison with CT imaging, ultrasound imaging is much safer, rapid, and patient-compliant. Furthermore, from the instrumentation point of view, ultrasound imaging machines are much smaller, inexpensive, and can be easily converted into mobile devices for point-of-care applications. In a recently concluded study, Bianchi and co-workers assessed the potential of LUS as a point-of-care set-up for the early detection of COVID-19 in a hospital in Italy [124]. Remarkably, the LUS imaging demonstrated a sensitivity of 97% and could accurately detect COVID-19 with 97% positive predictive value (PPV). In a similar study, Gioia and co-workers tried to predict COVID-19 pneumonia from LUC imaging [125]. Using LUC data on 250 patients, they achieved a sensitivity of 85% and specificity of 91% for the detection of COVID-19 pneumonia. The overall positive and negative predictive values were 86% and 91%, respectively. In another LUS study, Copetti et al. reported a diagnostic accuracy of 98.7% for the detection of COVID-19 in health care workers [126]. Although LUS imaging offers several advantages over CT imaging and RT-PCR-based diagnosis, further studies are required on larger data set before it can be considered for clinical purposes. Nonetheless, LUS imaging could provide a supporting test alongside CT and RT-PCR for the early detection of severe COVID-19 cases and thus could help in taking timely preventive measures.

## 3. Other Diagnostic Techniques

### 3.1. Biosensors

Biosensors are detection devices that rely on the specific interaction of the whole virus or viral molecules with a biorecognition element (e.g., antibody, oligonucleotide, peptide, etc.) immobilized on the surface or in the vicinity of a transducing element. The transducing element converts this interaction into a measurable optical or electrical signal. The cost-effectiveness, portability, scalability, and simple operative procedure make biosensors attractive for the mass-level diagnostic screening of the population. Several different types of biosensors are currently being developed. These biosensors have huge potential for rapid SARS-CoV-2 detection, but are yet to receive approval from regulatory authorities. A few of the promising biosensors are discussed. Some of the diagnostic techniques available for use in clinical laboratories and in research are discussed in Table 1.

#### 3.1.1. Colorimetric Biosensors

To address the current global SARS-CoV-2/COVID-19 pandemic, mass-level testing is needed to overcome this situation. In view of this, the colorimetric testing technique is very easy and simple to perform, and the outcomes of the experiment can be identified by the naked eye in terms of qualitative or quantitative results [127,128]. This biosensor works on the principle in which color change is detected, and it is linked to specific biochemical reactions between the analyte and biosensing receptors on the colorimetric biosensor’s surface. The detection of color change can be conducted simply without sophisticated devices, and it does not require an expert person. Nanoscience and nanotechnology play an important role in detection, and it has manifested tremendous advantages in colorimetric assays. AuNP is suitable a nanomaterial for colorimetric sensing due to its attractive optical properties, tunable size, and surface plasmon. The sensitivity of the assay can be also enhanced by using nanoparticles, such as colloidal gold nanoparticles (AuNPs). In the case of SARS-CoV-2 virus detection, virus surface antigens are mostly targeted with specific bioreceptors (e.g., antibody/nucleic acid). Once the reaction between the viral protein antigen and antibody takes place, the specific color on the strip of the sensor is developed [129]. Moreover, the color generated on the strip can be visualized via the naked eye, or it can also be sensed by detectors present in sensors. There are many companies that have used AuNPs to increase the sensitivity of lateral flow assay-based colorimetric immunosensor/biosensors [130,131].

#### 3.1.2. Plasmonic Biosensors

These biosensors exploit the extraordinary plasmonic properties of gold, silver, or copper nanomaterials to construct colorimetric, optical, or spectroscopic optical biosensors. Several plasmonic biosensors for the detection of viral pathogens have been reported previously, which has laid a solid foundation for constructing a diagnostic biosensor for COVID-19. In this direction, a dual-function plasmonic–photothermal biosensor for the detection of SARS-CoV-2 directly from an air sample was reported by Wang and co-workers [132]. The specific recognition of SARS-CoV-2 was achieved with oligonucleotides complementary to RNA. Regarding specific hybridization, the local surface plasmon resonance (LSPR) signals from the gold nano-islands (AuNIs) quantitatively varied with the viral load. Interestingly, the thermo-plasmonic property of AuNIs was also coupled to increase the temperature for in situ hybridization, which significantly improved the specificity of the biosensor. In another study, the LSPR property of dispersed gold nanoparticles was used to fabricate a colorimetric diagnostic assay for SARS-CoV-2 [133]. The color change was brought about by the aggregation of colloidal gold nanoparticles on a specific hybridization of immobilized oligonucleotides with viral RNA, as portrayed in Figure 4. The test could be completed within 10 min of viral RNA isolation and had a reported LOD of 0.18 ng/µL. Interestingly, the development of a field-deployable, hand-held, low-cost, plasmonic fiber–optic absorbance biosensor (P-FAB) was proposed for the detection of the SARS-CoV-2 virus from patient saliva samples. Specific identification of the virus was achieved with immobilized antibodies against the N-protein of the virus [134]. Recently, Calvo-Lozano and co-workers reported an in-house-developed microfluidic SPR biosensor for the label-free detection of antibodies against SARS-CoV-2 RBD and N-proteins in serological samples [135]. The SPR sensor was arranged in the Krechmann configuration, and the antigen–antibody binding was monitored in real-time through the shift in the resonance wavelength. During antibody binding, the resonance wavelength red-shifts due to the increasing refractive index. The developed SPR sensor demonstrated a remarkable sensitivity of 99% with 100% specificity, and was clinically validated. A similar SPR system capable of detecting antibodies against the SARS-CoV-2 S-protein and N-protein was reported by Basso and co-workers [136]. These studies clearly demonstrate the excellent potential of plasmonic biosensors in clinical diagnosis for the label-free, sensitive, and specific detection of clinical analytes in the near future.

#### 3.1.3. Electrochemical Biosensors

Electrochemical biosensors are another promising platform for developing low-cost, rapid, and robust diagnostic tools for COVID-19 [137]. The huge market presence of electrochemical glucose biosensors already affirms their potential in this regard. Apparently, some electrochemical biosensors are already in the development phase and awaiting the necessary approval to undergo clinical studies. For example, the US biotech firm Nano DiagnosiX is developing a rapid-detection test for SARS-CoV-2 biosensing on their existing femtospot biosensor chip (https://www.nanodiagnosix.com/ (accessed on 14 March 2022)). The biosensor utilizes a nanoribbon field-effect transistor (FET) chip to monitor the minute electrochemical change upon analyte binding. Viral protein or complementary oligonucleotides could be used for the selective identification of the SARS-CoV-2 virus or neutralizing antibodies in the patient sera. In another study combining highly conductive graphene with FET (gFET), an ultrasensitive biosensor for the detection of SARS-CoV-2 was reported by Seo et al. [138]. The biosensor was fabricated by immobilizing antibodies against SARS-CoV-2 spike proteins on the graphene’s surface. It could detect spike proteins at a concentration of 1 fg/mL in phosphate-buffered saline (PBS) solution and 100 fg/mL clinical transport medium used for sampling. Interestingly, the sensor could also sensitively detect the whole virus obtained from cultured cells with a LOD of 2.42 × 10^2^ copies/mL. In a unique approach, Mavrikou et al. reported a portable, ultra-rapid, and ultrasensitive biosensor for the detection of the SARS-CoV-2 virus based on the bioelectric recognition assay (BERA) [139]. This is a whole-cell-based biosensing assay wherein the electric response of cell receptor–analyte binding is quantified. Consequently, to sense the virus particles, Mavrikou et al. created membrane-engineered mammalian cells (Green monkey kidney cells) by incorporating the membrane surface with human chimeric anti-spike S1 protein antibodies. Upon interaction with viral particles, the cells produced a selective, quantitative, and reproducible electrical response due to the change in membrane potential and other electrical properties of the cells. The assay exhibited ultrasensitive biosensing capability with a limit of detection of 1 fg/mL and total detection time of just 3 min. In a similar approach, Rafailovich and co-workers utilized a three-dimensional molecular imprint of a SARS-CoV-2 virion on a self-assembled monolayer (SAM) of thiols on a gold chip [140]. The molecular imprint acted as an exact inverse replica of the virus particles and facilitated specific binding to the gold’s surface. The specific binding was then determined by measuring the open-circuit potential (OCP) of the substrate. Using this approach, the sensor was able to detect whole virions (H1N1 and H3N3 Influenza) and isolated surface proteins of the SARS-CoV-2 and Middle Eastern Respiratory Syndrome (MERS) viruses in human saliva. The sensors also exhibited remarkable limits of detection (LODs) of 200 PFU/mL and 100 pg/mL viral particles and surface spike proteins, respectively.

In a similar whole-cell-based biosensing format, Smith Detections Inc. announced Biosflash^®^ for the rapid detection of airborne SARS-CoV-2 viral particles (https://www.smithsdetection.com/helping-detect-covid-19/ (accessed on 14 March 2022)). The sensor works on the CANARY (cellular analysis and notification of antigen risks and yields) principle developed previously. It utilizes proprietary, genetically engineered immune cell-expressing pathogen-specific antibodies at the surface. Upon positive interaction with the pathogen, the engineered cell produces a fluorescent signal that could be quantified [141].

In a true sample-to-answer format, GenMark Diagnostics Inc., US, developed a proprietary microfluidic-based ePlex SARS-CoV-2 electrochemical testing setup. The device uses the DNA hybridization assay for capturing reverse-transcribed and PCR-amplified viral cDNA from viral RNA on a gold electrode. A second ferrocene-labeled reporter probe is then used to generate a quantitative electrochemical signal through voltammetry, as shown in Figure 5. The device takes around 50 min of reporting time for the completion of the whole process, beginning from sample processing to quantitative result information. The device has a reported sensitivity of 94.4% with a LOD of 10^5^ copies/mL. The device has been recently approved for qualitative COVID-19 diagnosis by the US Food and Drug Administration (US FDA) under emergency use authorization (EUA) (the EUA is available at https://www.fda.gov/media/136282/download (accessed on 14 March 2022)).

Using an innovative approach, Lee and co-workers developed a multiplex electrochemical biosensor that can simultaneously identify two target genes (S and Orf1ab genes) of the SARS-CoV-2 virus within 1 h. The detection is highly specific and sensitive, with the limit of detection for both targets in the <10 ag/µL (10–18 g) range. Incredibly, such low-level detection is achieved without using any nucleic acid amplification step, which significantly lowers the operational cost as well as encourages the development of a true point-of-care diagnostic device [142]. In another unique approach, Reuel and co-workers demonstrated a highly sensitive inductor–capacitor (LC) resonator-based electrochemical sensor for the detection of SARS-CoV-2 [143]. This unique approach utilizes a gelatin-switch resonator (GSR) as the transducing element and a toehold switch linked with a cell-free expression system against the N-gene of SARS-CoV-2 as the biorecognition element. Upon hybridization with the target N-gene of SARS-CoV-2, the toehold switch relaxes and induces the cell-free expression of the protease enzyme, which, in turn, degrades gelatin, thus changing the frequency of the LC resonator. The toehold switch RNA provides exceptional specificity, allowing the selective detection of SARS-CoV-2 among other seasonal coronaviruses, as well as SARS-CoV-1. The limit of detection of the device was reported to be 100 copies/µL.

In a unique proof-of-concept study, Snitz et al. attempted to detect and identify COVID-19-positive cases from the volatile organic compounds (VOCs) exhaled by suspected patients [144]. For this purpose, they utilized the commercially available eNose (PEN3, AIRSENSE Analytics GmbH, Schwerin, Germany), which can measure several types of VOCs emitted in a person’s body odor or exhaled breath (in this case). The data of VOC measurements from more than 500 participants were then subjected to computational analysis using deep learning classifiers. Interestingly, the authors reported a significantly high true-positive rate of 66.7% for the identification of COVID-19 from the exhaled breath of the participant. The positive identification rate was even higher for non-symptomatic participants, at >75%. However, the algorithm also reported a high false-positive rate of around 57%, which limits its application as a diagnostic tool. Nonetheless, considering the rapidity and feasibility of the test and the almost-instant results, these types of tests offer substantial advantages over other molecular diagnosis or serological tests [145,146,147].

### 3.2. Spectroscopic Biosensors

Unlike traditional optical (fluorescent and chemiluminescent), colorimetric, or electrochemical biosensors, spectroscopic techniques for biosensing offer tremendous advantages, such as ultrafast results, little or no sample preparation, multiplexing, on-site operability, and large-scale and real-time sample analysis capability [148]. Such properties are particularly suitable in the current situation, wherein the transmission of SARS-CoV-2 could be effectively prevented by large-scale population screening and isolating the affected population quickly [149].

Researchers from the Catalan Institute of Nanoscience and Nanotechnology, Barcelona, Spain, recently announced the development of a spectroscopic biosensor called CONVAT, which is based on bimodal wave interferometry (CoNVat Project, Grant ID 101003544, funded under Horizon 2020 No. H2020-EU.3.1.3.). The basic principle behind the sensing technique includes the capturing of SARS-CoV-2 virus on the sensor’s surface via immobilized captured antibodies or complementary oligonucleotides to viral RNA. This recognition event causes a change in the refractive index of the sensing region and thus minutely shifts the direction of light propagation, which can then be quantified (Figure 6). The very high sensitivity of the signals is expected to detect SARS-CoV-2 in the 10^−12^ to 10^−18^ M range. Another promising spectroscopic biosensor for COVID-19 diagnosis is currently under development by Yeh and his co-worker [150]. The microfluidic biosensor named VIRRION (virus capture with rapid Raman spectroscopy detection and identification) is capable of capturing and enriching viral particles through differential filtration porosity created from a network of carbon nanotube (CNT) arrays. The captured virus is then identified in real time by surface-enhanced Raman spectroscopy (SERS). The biosensor could successfully identify human influenza virus within a few minutes of detection time and reported a detection limit of 10^2^ EID50/mL (50% egg infective dose per microliter) [150]. The biosensor is currently being configured for the detection of the SARS-CoV-2 virus. Recently, several SERS-based identification techniques for SARS-CoV-2 has been reported [151,152]. These detection methods offer simple, rapid, and reagent-free detection of SARS-CoV-2 virus particles in patient saliva. Aided by computational analysis techniques, such as deep learning, SERS-based detection has achieved remarkable sensitivity and specificity, which can even rival that of the gold standard RT-PCR [149].

**Table 1 vaccines-10-01200-t001:** Comparative analysis of different diagnostic strategies used for the diagnosis of SARS-CoV-2.

Name of Detection Technique	Type of Test	Infrastructure	Time of Detection	Accuracy	Qualitative OR Quantitative	Time Required after Infection for Detection	Medical Use	Reference
Reverse-transcription-polymerase chain reaction (RT-PCR)	Nucleic acid-based	Laboratory setup needed	4 h	High	Both types	1–8 days	Mostly used test	[49,50,53]
Isothermal nucleic acid amplification	Nucleic acid-based	Laboratory setup needed	>3 h	High	Both types	1–8 days	Moderately used test	[67]
Reverse-transcription loop-mediated isothermal amplification (RT-LAMP)	Nucleic acid-based	Laboratory setup needed	30 min^−^^1^ h	High	Both types	1–8 days	Small use	[69,70,71]
Transcription-mediated amplification (TMA)	Nucleic acid-based	Laboratory setup needed	30 min	High	Both types	1–8 days	Small use	[76,78]
Rolling circle amplification	Nucleic acid-based	Laboratory setup needed	90 min	High	Both types	1–8 days	Small use	[79,80]
CRISPR-based assays	Nucleic acid-based	Laboratory setup needed	1 h	High	Both types	5–10 days	Moderately used test	[82,83,84]
Nucleic acid hybridization microarray assays	Nucleic acid-based	Laboratory setup needed	2 h	High	Only quantitative	5–10 days	Not in use: research stage	[90,91,92]
Nucleic acid sequencing	Nucleic acid-based	Laboratory setup needed	2 h	High	Only qualitative	5–10 days	Moderately used test	[95,97,98]
Enzyme-linked immunosorbent assay (ELISA)	Antibody detection	Laboratory setup needed	2 h	High	Both	5–10 days	Moderate	[110,111,112]
Lateral flow assay (LFA)	Rapid antigen detection	Point-of-Care Device	10–30 min	High	Moderate to high	10 days	Moderately used test	[115,117]
Luminescent immunoassay	IgM and IgG antibody detection in patients’ serum	Laboratory setup needed	2–3 h	High	Both	5–10 days	Moderately used test	[118,119,120]
Protein microarray	IgM and IgG antibody detection	Laboratory setup needed	20–30 min	Moderate	Both	5–10 days	Moderate to high	[121]
Agglutination assay	Antibody detection	Point-of-Care Device	10–30 min	Moderate	Only qualitative	5–10 days	Moderately used test	[122,123]
Chest computed tomography (CT) scan	Imaging-based diagnosis—radiography	Complex and costly test	4–5 h	High (if lung infected)	Both	4–15 days	Second after RT-PCR	[128,129]
Lung ultrasound (LUS)	Imaging-based diagnosis	Point-of-Care Device (portable device)	1 h	High	Both	4–15 days	Moderately used test	[131,133]
Plasmonic biosensors	Detection of viral pathogens	Point-of-Care Device	10–30 min	Moderate	Both	10 days	Not in use: research stage	[133,134]
Electrochemical biosensors	Antigen detection (SARS-CoV-2 spike proteins)	Point-of-Care Device	30–60 min	Moderate	Both	10 days	Not in use: research stage	[141,142,146,147]
Spectroscopic biosensors	Detection of viral pathogens/RNA	Point-of-Care Device	10 min	Moderate	Both	10 days	Not in use: research stage	[150,151]

Researchers from the Ben–Gurion University of the Negev (BGU), Israel, have announced a groundbreaking technology based on terahertz resonance spectroscopy that could, in real time, detect the SARS-CoV-2 virus from just a simple breath analysis. The technique can identify affected individuals within as little as 20 s and could be used as a vital installation in airports, offices, and marketplace for the screening of people. Presently, the device is under the validation phase and is expected to hit the market as early as September this year (https://www.eurekalert.org/pub_releases/2020-05/aabu-ome052220.php (accessed on 14 March 2022)).

Infra-red spectroscopy is another highly promising spectroscopic technique that has been extensively explored in the past for medical diagnosis [153,154,155]. In line with other spectroscopic techniques, infra-red spectroscopy also provides rapid, real time, reagent-free, and highly sensitive detection. Recently, a number of studies have been published, wherein IR spectroscopy, aided by computational analysis, has been utilized to selectively identify SARS-CoV-2 infection in suspected patients [156,157]. Nogueira and co-workers utilized attenuated total reflection (ATR) Fourier-transform infrared (FTIR) spectroscopy to detect SARS-CoV-2-positive samples among 243 oropharyngeal swab suspensions [156]. Subsequent data analysis with computational techniques achieved a sensitivity of greater than 80% along with a moderate specificity of ~65%. In contrast, the method developed by Kitane et al. achieved a remarkable sensitivity and specificity of 97% and 98.3%, respectively [157]. The achieved sensitivity and specificity of the method are very close to those of conventional RT-PCR and could potentially be adopted in clinical practices. Such significant differences in spectroscopic measurement can occur through variations in the sample type, measurement scheme, and even computation algorithms utilized for analysis. Thus, it is essential to continuously reiterate spectroscopic measurements on a large sample size so that intra-sample variations can be minimized.

### 3.3. Piezoelectric Sensors

A piezoelectric sensor is a label-free biosensor device that works on variations in oscillation frequency upon biorecognition reaction [158]. This sensor is based on the piezoelectric effect, and it uses piezoelectric materials that can produce a voltage once they are mechanically stressed. Piezoelectric biosensors are very sensitive and can even detect femtograms of mass change. The piezoelectric crystal’s surface is coated with bioreceptors (antibody/peptide/aptamer etc.), and it interacts with the analyte and consequently reduces the oscillation frequency of the quartz–crystal material [159,160]. This biosensing technique is also practiced in the diagnosis of different pathogens in biological samples, as represented in Figure 7. In this existing, worldwide pandemic situation, COVID-19 has spread nearly everywhere. In such a circumstance, this biosensor is a tempting tool, as it does not require any sophisticated laboratory facility and skill [161].

### 3.4. Microfluidic Sensor

Microfluidics refers to technology that involves the fabrication of microminiaturized channels and chambers in order to allow small volumes of fluids to flow. Recently, there has been an incredible rise in the combination of sensing devices with microfluidics systems. Microfluidic sensors offer numerous advantages, such as small sample volumes, rapid detection times, high portability, and ease of multiplexing, that make them appropriate for biosensor development applications. In this technique, small amounts and precise volume control give rise to the miniaturization of lab-on-a-chip diagnostics for the application of biosensors [162,163]. These point-of-care (PoC) devices can offer the detection of analytes and may deliver rapid diagnosis tools to patients at onsite and remote regions with limited health care systems [164]. Lillehoj and his group developed a microfluidic chip to detect the SARS-CoV-2 nucleocapsid protein by using dual-labeled magnetic nanobeads for immunomagnetic enrichment and signal amplification [165]. This sensing platform offers minimum sample and reagent consumption, enhanced detection sensitivity, and easy handling. The microfluidic sensor platform has exhibited highly sensitive and specific detection capacity in serum samples, with a limit of detection (LOD) of about 50 pg/mL in the whole serum within 1 h and 10 pg/mL in five-times-diluted serum within half an hour. To make it portable, the sensor is integrated with smartphones using potentiostat that does not require any external power supply. In another study, Hwang-soo et al. introduced a different method to detect the COVID-19/SARS-CoV-2 virus through the isothermal amplification of nucleic acids by means of applying a mesh with multiple pores [166]. In this bio-sensing platform, the blockage of pores takes place via the establishment of DNA hydrogel once the hybridization of the SARS-CoV-2 virus and probe DNA occurs.

## 4. Conclusions and Future Outlook

After several years, the whole world experienced a medical emergency created by the SARS-CoV-2 virus due to its high infection rate. The SARS-CoV-2/COVID-19 pandemic has drawn attention globally to how novel viral pathogens can rapidly emerge and spread through the human population and, in due course, cause severe public health emergencies. To understand the pathogenicity of this novel viral pathogen, basic and clinical research should be conducted to investigate its replication, transmission dynamics, and pathogenesis in humans. These scientific and clinical studies may help to develop and evaluate potential diagnosis and therapeutic strategies against the SARS-CoV-2/COVID-19 pandemic. Moreover, existing trends suggest the occurrence of future outbreaks of SARS-CoV-2 due to changes in the climate, and environmental conditions could be linked with human–animal contact. On other hand, the seafood market at Huanan City, situated in South China, symbolizes the ultimate conditions for interspecies contact of wildlife with domestic animals, such as birds, pigs, and mammals. This significantly upsurges the possibility of interspecies transmission of SARS-CoV-2 infections and can result in high risks to human populations due to adaptive genetic recombination in these viruses.

The current pandemic caused by SARS-CoV-2 will be contained shortly, just like previous outbreaks. However, the factual inquiry is how are we preparing to tackle the next zoonotic viral outbreak that is likely to occur within the next few years, or maybe sooner? Our scientific knowledge of most zoonotic viral infections is limited. Moreover, these viruses have not been isolated and studied by the scientific community. Therefore, extensive studies must be conducted on such viruses to overcome this type of viral pandemic situation once they are related to specific disease epidemics. To evaluate the risk of future epidemics, it is urgently needed to conduct both in vitro and in vivo studies by using appropriate animal models. Presently, approved antiviral medications and vaccines against SARS-CoV-2 exist, with limited availability. However, improvements in designing vaccines and antiviral medicines against various other evolving diseases will offer a way to develop suitable therapeutic agents against any virus and pathogen, such as COVID-19, in a short period.

## Figures and Tables

**Figure 1 vaccines-10-01200-f001:**
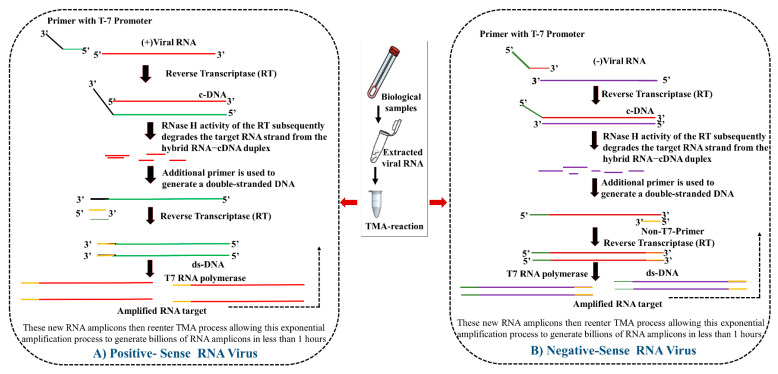
Schematic presentation of the transcription-mediated amplification (TMA) of viral RNA (**A**) Positive-sense RNA virus; (**B**) Negative-sense RNA virus. In general, the promoter sequence of the T7 RNA polymerase uses the dsDNA as a template, generates a new RNA sequence, and produces the same DNA copy number.

**Figure 2 vaccines-10-01200-f002:**
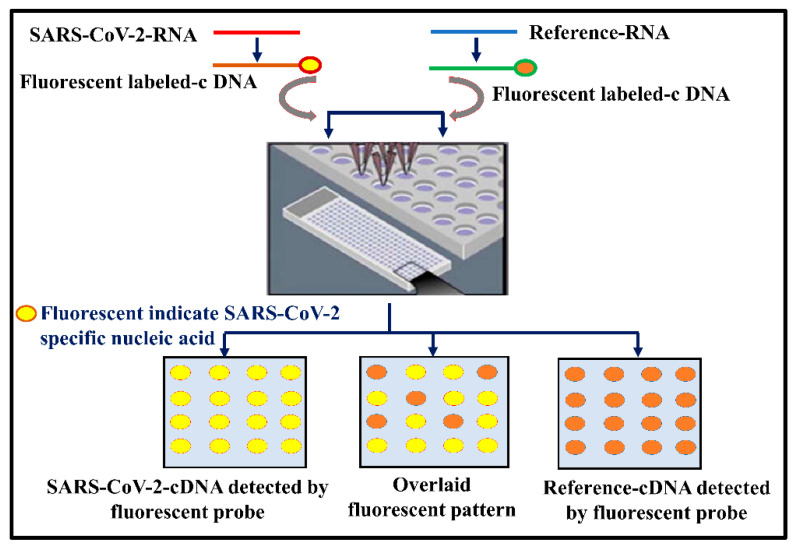
Nucleic acid hybridization using microarray. Viral cDNA and reference cDNA with different fluorescent labels are mixed and applied to microarray wells coated with specific DNA probes.

**Figure 3 vaccines-10-01200-f003:**
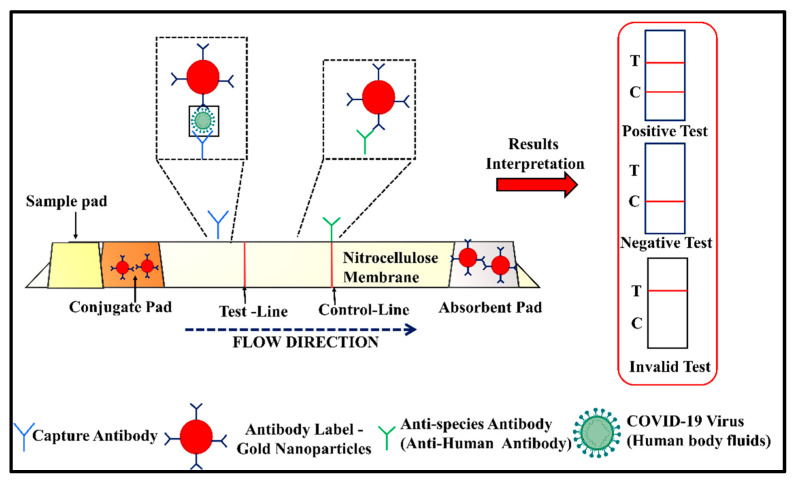
Graphic illustration of a lateral flow immunoassay (LFA) for the detection of the COVID-19 virus. Immobilized viral antigens and, if the patient sample contained any, antiviral antibodies pass over it through capillary action, forming a complex that is colorimetrically detected.

**Figure 4 vaccines-10-01200-f004:**
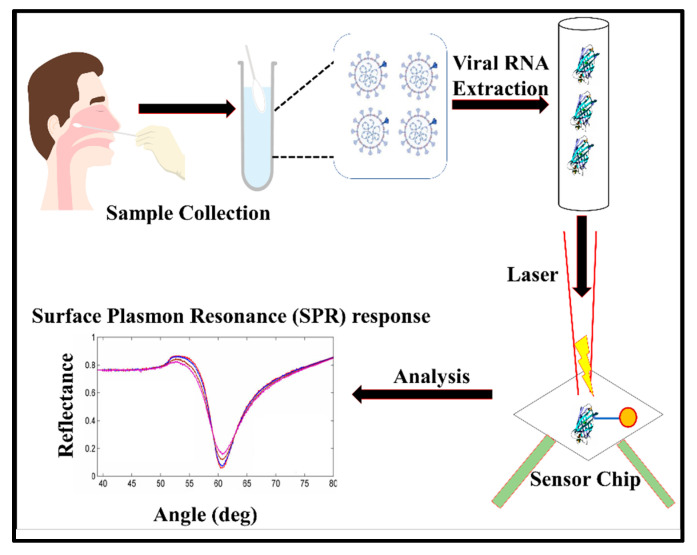
Diagrammatic presentation of the detection of COVID-19 virus RNA by surface plasmon resonance (SPR) using a gold nanoparticle-labeled probe (antisense oligonucleotides).

**Figure 5 vaccines-10-01200-f005:**
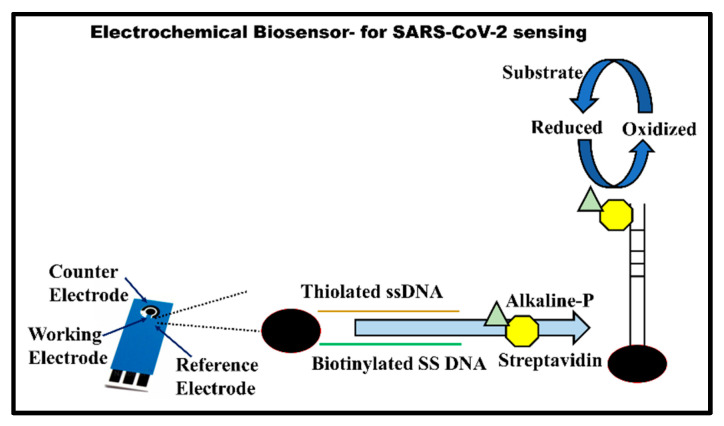
Diagrammatic presentation for the detection of COVID-19 virus RNA by an electrochemical biosensor. The device uses hybridized DNA for capturing reverse-transcribed and PCR-amplified viral cDNA from viral RNA on a gold electrode.

**Figure 6 vaccines-10-01200-f006:**
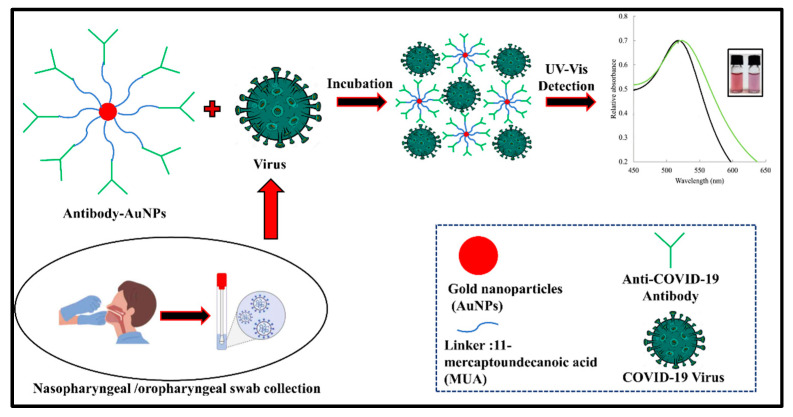
Graphic representation of a colorimetric sensor for SARS-CoV-2 detection. Anti-SARS-CoV-2 antibodies conjugated with AuNP form a complex with the virus upon incubation with a sample containing SARS-CoV-2 viral particles. UV-vis detection of the SARS-CoV-2 virus based on the aggregation of AuNP-conjugated anti-SARS-CoV-2 antibodies on the SARS-CoV-2 spike antigen’s surface.

**Figure 7 vaccines-10-01200-f007:**
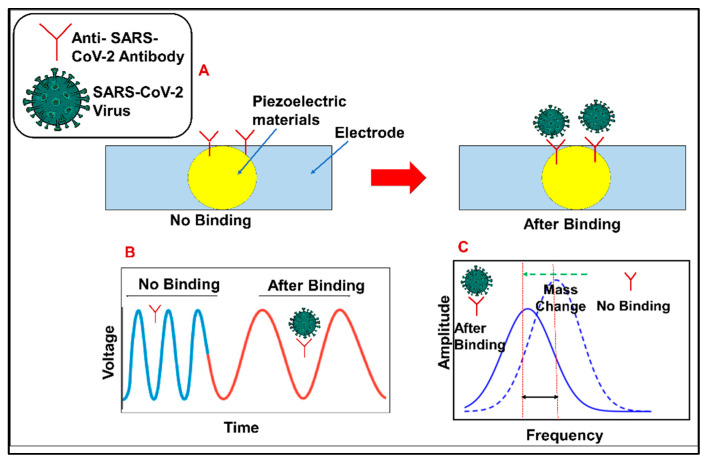
Pictorial representation of piezoelectric sensors for SARS-CoV-2 detection. (**A**) Piezoelectric sensor detecting the SARS-CoV-2 viral antigen. Due to the adsorption of the virus on piezoelectric materials, virus-antibody interaction resultant-on-mass changes lead to sensor bending, which changes the signal. (**B**) Diagram of voltage against time and (**C**) amplitude against frequency.

## Data Availability

Not applicable.

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
