# Peer review of "Diagnostic Tools for Rapid Screening and Detection of SARS-CoV-2 Infection"

_vaccines, 2022, doi:10.3390/vaccines10081200_

Round 1
Reviewer 1 Report
I appreciate the authors' effort to revise the manuscript vaccines- 1721111. The manuscript has improved significantly with the reviewers’ recommendations.
The authors have addressed most of the comments and suggestions.
The references have been conveniently updated.
However, several aspects still need to be considered:
· The number of figures continues to be scarce. Additional figures could help to explain the fundamentals and applications of the proposed techniques.
· The Table containing the key characteristics of each diagnostic strategy could contribute to summarizing the benefits and drawbacks of the methods.
· The recommendation concerning the biosensors section has not been sufficiently addressed. Since some of the applications are based on a serological approach, this fact should be conveniently clarified in the text.
Author Response
REVIEWER I
I appreciate the authors' effort to revise the manuscript vaccines- 1721111. The manuscript has improved significantly with the reviewers’ recommendations.
Thank you so much for appreciating our efforts.
The authors have addressed most of the comments and suggestions.
Yes, we have accepted most of your advice/suggestions and done the changes in the manuscript (MS) accordingly. Please read out and let us know your opinion. Thank you in advance.
The references have been conveniently updated. However, several aspects still need to be considered:
- The number of figures continues to be scarce. Additional figures could help to explain the fundamentals and applications of the proposed techniques.
In the revised version we have added two more figures (figure # 6 and 7) for spectroscopic and piezoelectric sensors for SARS-CoV-2 detection. Please find them in their respective sections 3.2 and 3.3
- The Table containing the key characteristics of each diagnostic strategy could contribute to summarizing the benefits and drawbacks of the methods.
We have summarized all the described diagnostic methods/strategy being use for SARS-CoV-2 detection in the form of Table. We have compared each of them based on accuracy, time of detection, sensitivity etc. Please find Table 1 in page number 15 and 16.
- The recommendation concerning the biosensors section has not been sufficiently addressed. Since some of the applications are based on a serological approach, this fact should be conveniently clarified in the text.
In the revised version of MS, we have incorporated more information with latest references from the biosensors field. Please find them in the section 3.1.1, 3.3 and 3.4 in page number 16,21 and 22 in the revised MS.
- Ember, K.; Daoust, F.; Mahfoud, M.; Dallaire, F.; Zamani, E.; Tran, T.; Plante, A.; Diop, M.-K.; Nguyen, T.; St-Georges-Robillard, A.; et al. Saliva-based detection of COVID-19 infection in a real-world setting using reagent-free Raman spectroscopy and machine learning. Journal of Biomedical Optics 2022, 27, 025002.
- Kumar N, Shetti NP, Jagannath S, Aminabhavi TM. Electrochemical sensors for the detection of SARS‐CoV‐2 virus. Chem Eng J. 2022, 430:132966.
- Sharma, S.; · Shrivastava, S.; Kausley, S.B; Rai, B.;Pandit, A B.Coronavirus: a comparative analysis of detection technologies in the wake of emerging variants. Sensors Internationa, 2022, 26,1-19. https://doi.org/10.1007/s15010-022-01819-6.
- Singh,B.; Brateen Datta, B.; Ashish, A.; Dutta, G.A comprehensive review on current COVID-19 detection methods: From lab care to point of care diagnosis. Sensors Internationa 2021, 2,100119. https://doi.org/10.1016/j.sintl.2021.100119.
Please find here revised version for your further evaluation. Thanks

Reviewer 2 Report
New text was added to the manuscript, however, the previous suggestions for revision were not properly attended. The title remains with the same meaning as before, despite the broad topic the review process could be displayed.
Author Response
REVIEWER 2
New text was added to the manuscript, however, the previous suggestions for revision were not properly attended. The title remains with the same meaning as before, despite the broad topic the review process could be displayed.
I would like to thank to the reviewer for constructive comments which makes the MS more consolidated and informative. In the present format of manuscript, we have changed the title as well as only relevant information have been discussed. Please see the modified manuscript as per the suggestion and let us know your opinion. Thank you in advance.

Reviewer 3 Report
Dear authors,
I found your work very interesting. I appreciated it. You gave a wide overview of the diagnosis of Sars-CoV2 virus but in general of all virus.
I have only some minor point regarding orthography:
- line 531: you wrote "are can", I guess you should delate "are"
-line 646: you wrote "are could2, I guess you should delate "are"
-line 706: you wrote "an" but it is "a"
- line 935: you wrote "and" but I guess you have to delate it.
Author Response
REVIEWER 3
Dear authors,
I found your work very interesting. I appreciated it. You gave a wide overview of the diagnosis of Sars-CoV2 virus but in general of all virus.
Thank you for acknowledging our efforts.
I have only some minor points regarding orthography:
We apologize for these minor mistakes left there. We have now proofread the manuscript with full attention.
- line 531: you wrote "are can", I guess you should delate "are"
Correction has been made.
-line 646: you wrote "are could2, I guess you should delate "are"
Correction has been made.
-line 706: you wrote "an" but it is "a"
Correction has been made.
- line 935: you wrote "and" but I guess you have to delate it.—
We have made these corrections. Sorry that we have overlook these minor mistakes. Thank you for the suggestions which make the manuscript more reader friendly. Please read out the modified manuscript and let us know your opinion.
Thank you in advance.

Reviewer 4 Report
The manuscript by Pandey et al is a comprehensive review of diagnostic tools for the SARS-CoV2 disease,and was an enjoyable and informative read. The review is well organized and describes a huge volume of research. As a likely consequence of this, it was somewhat superficial in places. However, this is overcome by the reasonable number of references cited. Notwithstanding, I did find that the authors missed some published work that can add to the review:
- A pertinent electrochemical work that draws a comparison between virial detection methods (10.1016/j.cej.2020.127575);
- A work that assesses the techniques available/described, tackling on relative advantages and interpretation of results (10.1039/D0AN01686A);
- A fascinating reagent-free electrochemical sensing detection method.
It is my believe that this three works should be referenced.
Given the length of the review, there are relatively few figures, which makes it very text heavy. Additionally, not sure if is the pdf file available to review but picture quality is not all ok. For example, figures 1, 2 and 4 are in less quality than figure 3. Figure 1 could benefit from being schematically vertical rather than horizontal.
Overall, the use of English needs some attention, but this did not obscure the meaning. There are some points the authors need to address:
1) Acronyms are repeated in text, For example LOD is defined in line 356 and then repeatedly redefined in lines 623, 630, 767, 804, and 838;
2) Line 96 – COVID-19 spelling.
3) Line 411 – spacing.
4) In the reviewer copy there are chunks of text in red. Not sure why.
Author Response
REVIEWER 4
The manuscript by Pandey et al is a comprehensive review of diagnostic tools for the SARS-CoV2 disease, and was an enjoyable and informative read. The review is well organized and describes a huge volume of research. As a likely consequence of this, it was somewhat superficial in places. However, this is overcome by the reasonable number of references cited. Notwithstanding, I did find that the authors missed some published work that can add to the review:
Thank you for praising and understanding our efforts. We have made the significant changes in MS and make the manuscript more reader friendly by adding more figures and comparative table.
- A pertinent electrochemical work that draws a comparison between virial detection methods (10.1016/j.cej.2020.127575);
- A work that assesses the techniques available/described, tackling on relative advantages and interpretation of results (10.1039/D0AN01686A);
- A fascinating reagent-free electrochemical sensing detection method.
It is my believe that this three works should be referenced.
We have added these references as they are worth to mention here in our review. Sorry for overlook.
Given the length of the review, there are relatively few figures, which makes it very text heavy. Additionally, not sure if is the pdf file available to review but picture quality is not all ok. For example, figures 1, 2 and 4 are in less quality than figure 3. Figure 1 could benefit from being schematically vertical rather than horizontal.
We have improvised the figure quality and make them uniform formatting. We are also submitting all the figure files separately for further suggestions. In the revised version we have added two more figures for spectroscopic and piezoelectric sensors for SARS-CoV-2 detection. Please find them in their respective sections 3.2 and 3.3. Additionally, we have summarized all the described diagnostic methods/strategy being use for SARS-CoV-2 detection in the form of Table. We have compared each of them based on accuracy, time of detection, sensitivity etc. Please find Table 1 in page number 15 and 16.
Overall, the use of English needs some attention, but this did not obscure the meaning. There are some points the authors need to address:
1) Acronyms are repeated in text, For example LOD is defined in line 356 and then repeatedly redefined in lines 623, 630, 767, 804, and 838;
Correction has been made.
2) Line 96 – COVID-19 spelling.
Correction has been made.
3) Line 411 – spacing.
Correction has been made.
4) In the reviewer copy there are chunks of text in red. Not sure why.
We have submitted the revised manuscript (via email) with clear and track changes (highlighted the changes) version separately. Consequently, red color text were the amendments, which we have done in the earlier version. Probably, you received the second file, I apologize for that.

Reviewer 5 Report
Overall Comment: The manuscript requires proof reading by someone well versed in English as there are many places where the intent or meaning is unclear and/or confusing.
Some areas of the manuscript need to be referenced better – see specific examples. This manuscript is unfocused in many areas, for example pgs 4 to 9 where many assay principles are described in depth but what is the relevance of this to the emerging diagnostic tools for detection ?
Several paragraphs in red font – not clear why ?
Title: The title suggests emerging techniques but the bulk of the manuscript also discusses assays that have “emerged”
Do you mean that new diagnostic tools can identify infected individuals sooner ? If so then the content of the article covers all the current tests and as well some that have potential in the future – some level of focus is needed for those assays which hold promise.
Abstract:
Lines 18 to 21 – not clear on the meaning. Would add that vaccination is also helpful in reducing the burden of infection and/or transmission
Introduction:
Lines 33-35 – reference or website to substantiate statement
Lines 56-58 – references required
Lines 70 – 93 – not clear why this content Sections 1.1 is in red font
Sections 1.1 & 1.2 & 2 can be significantly shortened and made more relevant to the theme of the manuscript
Diagnosis
Pgs 3 to 11- Individual methodology principles are exhaustively discussed, e.g., rt-PCR, TMA etc, whereas the title is “emerging diagnostic tools” and many of these assays have been in wide diagnostic usage in one form or another for many years.
Minor points
Line 188-189 there are no “typical COVID-19 symptoms’ many of these overlap with other resp infections which is why we have to test for alternate etiologies, unless it is solely SARS-CoV-2 circulating.
Lines 238-241 – ref 59 refers to detection of SARS-CoV-2 in plasma, in my experience a relatively rare occurrence in the usual COVID infections, maybe has limited utility in research applications ?
Lines 212 to 241 are in red font – why?
2.2 Serologic/immunologic assays
Line 545 – as we now know, the presence of IgG antibody does not confer long-term immunity from our current experience with the emergence of various sublineages of Omicron, that can re-infect cases previously infected with a prior variant of concern
2.3 Image Based Diagnosis
Lines 714 to 716 – most studies would support the molecular testing is most sensitive in the prodrome and acute onset whereas the radiologic changes follow after symptoms develop. As well in mildly symptomatic & asymptomatic cases it is possible that there are no changes - so how would CT scans be cost-effective and accurate ?
Conclusion:
It is unclear what points the authors were making as this section was fragmented and incoherent
References
Plentiful but absent in some areas when need to support the point
Author Response
REVIEWER 5
Overall Comment: The manuscript requires proof reading by someone well versed in English as there are many places where the intent or meaning is unclear and/or confusing.
We have taken the help of native English speaker for make the language correction.
Some areas of the manuscript need to be referenced better – see specific examples. This manuscript is unfocused in many areas, for example pgs 4 to 9 where many assay principles are described in depth but what is the relevance of this to the emerging diagnostic tools for detection?
We have changed the title and minimized the discussion of current diagnostic tools up to some extent and added other tools which could be useful for detecting SARS-CoV2 infection. As some of those already being used for other viral infections detection.
Several paragraphs in red font – not clear why?
We have submitted the revised manuscript (via email) with clear and track changes (highlighted the changes) version separately. Consequently, red color text were the amendments, which we have done in the earlier version. Probably, you received the second file, I apologize for that.
Title: The title suggests emerging techniques, but the bulk of the manuscript also discusses assays that have “emerged”
We agreed with reviewer’s comment hence have changed the title “Disease and Diagnostic Tools for Rapid Detection of SARS-CoV2 infection”
Do you mean that new diagnostic tools can identify infected individuals sooner? If so then the content of the article covers all the current tests and as well some that have potential in the future – some level of focus is needed for those assays which hold promise.
Please refer to the section 3 of MS we have included some contemporary test used for other viral infections and could be a bait for SARS-CoV2 infection.
Abstract:
Lines 18 to 21 – not clear on the meaning. Would add that vaccination is also helpful in reducing the burden of infection and/or transmission
We have changed the abstract according to the overall reviewer’s suggestion. Please go through and suggest.
Introduction:
Lines 33-35 – reference or website to substantiate statement
Lines 56-58 – references required
Lines 70 – 93 – not clear why this content Sections 1.1 is in red font
Sections 1.1 & 1.2 & 2 can be significantly shortened and made more relevant to the theme of the manuscript
We are glad that reviewer has made the keen observation and pointed out these mistakes. We have changed the introduction with supporting references.
Diagnosis
Pgs 3 to 11- Individual methodology principles are exhaustively discussed, e.g., rt-PCR, TMA etc, whereas the title is “emerging diagnostic tools” and many of these assays have been in wide diagnostic usage in one form or another for many years.
We have changed the title and minimized the discussion current diagnostic tools up to some extent and added other tools which could be useful for detecting SARS-CoV2 infection. As some of those already being used for other viral infections detection.
Minor points
Line 188-189 there are no “typical COVID-19 symptoms’ many of these overlap with other resp infections which is why we have to test for alternate etiologies, unless it is solely SARS-CoV-2 circulating.
Yes, we agreed with the reviewer's comment, so we have removed the “typical” word in the sentence and incorporated ‘suspected’ in the revised MS.
Lines 238-241 – ref 59 refers to detection of SARS-CoV-2 in plasma, in my experience a relatively rare occurrence in the usual COVID infections, maybe has limited utility in research applications ?
Yes, we totally agree with your concern. But Ana P Tedim and his coworkers recently proposed SARS-CoV-2 RNA was detected, using ddPCR and RT-qPCR, in 91% and 87% of ICU patients, 27% and 23% of ward patients and 3% and 3% of outpatients. They also claimed that RT-qPCR is able to detect 34/36 (94.4%) patients positive for viral RNA in plasma by ddPCR. So, we just used his research statement in the present review MS as it is mentioned in his published research.
Lines 212 to 241 are in red font – why?
We have submitted the revised manuscript with clear and track changes (highlighted the changes) version separately. Consequently, red color text were the amendments, which we have done in the earlier version. Probably, you received the second file, I apologize for that.
2.2 Serologic/immunologic assays
Line 545 – as we now know, the presence of IgG antibody does not confer long-term immunity from our current experience with the emergence of various sublineages of Omicron, that can re-infect cases previously infected with a prior variant of concern.
Yes, we agree with your concern that IgG antibody does not provide long-term immunity on the basis of current experience. We removed the long -term immunity part in the revised MS.
2.3 Image Based Diagnosis
Lines 714 to 716 – most studies would support the molecular testing is most sensitive in the prodrome and acute onset whereas the radiologic changes follow after symptoms develop. As well in mildly symptomatic & asymptomatic cases it is possible that there are no changes - so how would CT scans be cost-effective and accurate ?
Yes, we are agreeing with logic highlighted by the reviewer and hence removed those line from the MS.
Conclusion:
It is unclear what points the authors were making as this section was fragmented and incoherent
We have reframed the conclusion section. Thank you pointing out these logical and valuable mistakes. We have accepted those and made the correction and modification in the current revised version. Please read and let us know your opinion. Thank you in advance.
References
Plentiful but absent in some areas when need to support the point
We have added more relevant references and this section has been taken care of more cautiously.

Round 2
Reviewer 2 Report
The review process is not yet being shown, but the content is satisfactory.
Author Response
The review process is not yet being shown, but the content is satisfactory.
Answer - Thank you for acknowledging our effort. I appreciate the effort of reviewer for providing the valuable comments to make the manuscript more suitable and informative.
Reviewer 5 Report
Overall comment:
Although the title of this revised manuscript has been changed much of the content and order of the material remains the same.
I am unclear what the aim of this manuscript is and what the author’s focus is as I have listed some of my concerns relating to a few of the major themes:-
1> Is this a review of all accepted technologies for the diagnosis of COVID because it also includes assays in research. Would it be better just to focus on those research assays which show potential as they are not currently “in production” and therefore cannot be used for diagnosing COVID infections
2> If as the title suggests “rapid detection”, then serology has no place as it takes time to mount an immune response, which is inconsistent with the title. Serology can be used for epi studies, vaccine responses, discerning natural from vaccines responses or asymptomatic infection/reinfection, etc
3> Molecular and antigen assays are currently widely used for making a diagnosis, often fairly quickly, e.g. Cephid COVID Genexpert takes about 45 to 60 mins, but I see no mention here. My question is are you intending to make this manuscript a comprehensive review of just all rapid molecular assays ?
4> Antigen assays are rapid and relatively inexpensive but I see only scant reference to one or two antigen assays ? Should these not get more coverage in this manuscript if they have potential in making a rapid diagnosis ?
5> Is this a review of the principles of molecular assays that have the are potential to be rapid because there is extensive discussion of these assay principles?
6> There are also discussions in the manuscript as to the suitability of different sample types for the diagnosis of COVID. Here there is a distinction which sample types would be suitable or easy to collect for testing which depends upon whether the individual is ambulant or intubated. Most commonly nasopharyngeal, throat, nasal would be collected in the ambulant person compared with BALs or endotracheal samples in the intubated who would be severely ill. Saliva, gargles and breath samples have not gained much traction due to their poorer sensitivities in the current menu of available commercial tests. As well there are differences in viral titres between symptomatic vs asymptomatic patients and which assays are best applied either to mass screening or a targeted approach.
In summary the authors need to decide on a specific theme for their manuscript rather than present a pot-pourri of disconnected ideas to the reader which is why in its current version I cannot recommend it for acceptance.
Author Response
Thank you for acknowledging our revision. I appreciate the effort of reviewer has been made for providing the valuable comments to make the manuscript more suitable and informative. Please find the answers to the specific queries here.
Q1. Is this a review of all accepted technologies for the diagnosis of COVID because it also includes assays in research. Would it be better just to focus on those research assays which show potential as they are not currently “in production” and therefore cannot be used for diagnosing COVID infections
Answer: The rapidity and the extent with which SARS-CoV-2 virus was able to infect human beings was never witnessed in the modern-day history. In fact, the rate of infection was so high, that within a few months of its initial reporting, the COVID-19 disease (caused by SARS-CoV-2) was declared a global pandemic. To contain the further spread of the infection, it was imperative to develop rapid detection platforms for screening and isolating infected patients from the healthy ones. Consequently, several already existing platforms for detection of other viral disease were adapted for detection of SARS-CoV-2. These platforms were either based on detection of virus specific nucleic acids or detection of viral antigen/antibody in the suspected patient’s sample. In addition to adapting these traditional detection platforms, several imaging and spectroscopy-based method of detections as well as nanomaterial-based biosensors were also reported which showed huge potential for rapid and high-throughput analysis. This review article is written to introduce its reader to ever rapidly evolving detection techniques for SARS-CoV-2 infection, including both conventional and emerging methods but which have huge potential for high throughput analysis.
For traditional techniques, emphasis has been given on those techniques (both molecular and serology) which has been approved with EUA by US FDA. A comprehensive list of such techniques could be found in below mentioned web links.
FDA Approved Molecular Techniques
https://www.fda.gov/medical-devices/coronavirus-disease-2019-covid-19-emergency-use-authorizations-medical-devices/in-vitro-diagnostics-euas-molecular-diagnostic-tests-sars-cov-2
FDA Approved Serological Techniques
https://www.fda.gov/medical-devices/coronavirus-disease-2019-covid-19-emergency-use-authorizations-medical-devices/in-vitro-diagnostics-euas-serology-and-other-adaptive-immune-response-tests-sars-cov-2
In order to better reflect the objective of the review, we have now changed a title of the review as “Diagnostic Tools for Rapid Screening and Detection of SARS-CoV-2 infection” to better reflect its objective.
Q2. If as the title suggests “rapid detection”, then serology has no place as it takes time to mount an immune response, which is inconsistent with the title. Serology can be used for epi studies, vaccine responses, discerning natural from vaccines responses or asymptomatic infection/reinfection, etc.
Answer: Since no available treatment regime is available against the novel COVID-19 disease caused by SARS-CoV-2, the only effect strategy adopted by the healthcare providers around the world is to rapidly identify and isolate infected patients. Several conventional diagnostic techniques, both molecular (nucleic acid based) and serological (antibody-antigen interaction based) were adapted for detection of SARS-CoV-2 virus. Both molecular and serological detection platforms have their advantages and disadvantages in terms of sensitivity, specificity and rapidity. In general, molecular techniques generates highly specific and sensitive diagnosis in a fairly rapid time (1-3 hours for sample preparation to results), however they are not suitable for large sample analysis due to expensive reagents and laboratory-based set-up for sample preparation and analysis. On the other hand, serological test can give fairly accurate results with rapidity (10-15 min) and with minimal sample preparation. The serological tests can also be conducted on-site which gives it unique advantage over molecular techniques for large population screening. For these reasons, serological tests have been the first-choice of diagnosis for SARS-CoV-2 and therefore has been included in the review. However, to reflect the purpose of the review more accurately, the title has been modified as indicated in previous response.
Q3. Molecular and antigen assays are currently widely used for making a diagnosis, often fairly quickly, e.g. Cephid COVID Genexpert takes about 45 to 60 mins, but I see no mention here. My question is are you intending to make this manuscript a comprehensive review of just all rapid molecular assays?
Answer: The number of different products approved by FDA for EUA has been steadily rising as increasing amount of information are getting available on SARS-CoV-2 genome. Therefore, to keep the review article concise, only the detection technique (molecular or serological) has been highlighted in the review. Cephid COVID Genexpert is an RT-PCR technique targeting 3 viral genes (RdRP, N2, E gene) simultaneously. The RT-PCR molecular technique had been explained already in the review. As explained above, the objective of the review articles is to introduce its reader about various conventional and emerging detection platforms that has been adapted for SARS-CoV-2 detection. For the conventional platforms (molecular and serological), technologies approved by US FDA for EUA has been emphasized, while for emerging technologies (CT imaging, spectroscopic or biosensors), only those technologies which has potential for high-throughput analysis has been highlighted in the review.
Q4 Antigen assays are rapid and relatively inexpensive but I see only scant reference to one or two antigen assays? Should these not get more coverage in this manuscript if they have potential in making a rapid diagnosis?
Answer: Serological assays such as ELISA, LFA, Agglutination Assay and Protein Microarray are all adaptable technologies to detect either SARS-CoV-2 specific antibodies or viral proteins (as antigens) in the patient sera. The basic principle and examples of SARS-CoV-2 detection using these technologies has been included in the review article.
Q5 Is this a review of the principles of molecular assays that have the are potential to be rapid because there is extensive discussion of these assay principles?
Answer: All serological tests are ultimately based on antigen-antibody interactions taking place on different platforms to produce qualitative or semiquantitative results. However, unlike serological tests, the mechanism of molecular test is variable and therefore often requires detailed discussion.
Q6 There are also discussions in the manuscript as to the suitability of different sample types for the diagnosis of COVID. Here there is a distinction which sample types would be suitable or easy to collect for testing which depends upon whether the individual is ambulant or intubated. Most commonly nasopharyngeal, throat, nasal would be collected in the ambulant person compared with BALs or endotracheal samples in the intubated who would be severely ill. Saliva, gargles and breath samples have not gained much traction due to their poorer sensitivities in the current menu of available commercial tests. As well there are differences in viral titres between symptomatic vs asymptomatic patients and which assays are best applied either to mass screening or a targeted approach.
Answer: Since method of collection of samples is also one of the major factors which can influence the sensitivity and specificity of molecular tests, therefore, a brief discussion on various sample collection methods is also included in brief. Although a comparative assessment of various collection methods has not been made in the review. This is to keep the review article concise and in-line with the theme of the article.

This manuscript is a resubmission of an earlier submission. The following is a list of the peer review reports and author responses from that submission.
Round 1
Reviewer 1 Report
Thank you for summarizing the new diagnosis methods for COVID-19. I have two minor concerns. (1) There is a trade-off relation among the detection time, the detection sensitivity, and the complexity of apparatus and protocols. Please add some comments about this. (2) Whereas the detection of nucleocapsid proteins is easy, that of S1 proteins, which are important for binding with ACE2, is so difficult. Some researchers have attempted to detect the S1 proteins using ELISA and other methods. Please describe this issue in your review article.
Reviewer 2 Report
The manuscript viruses-1455131 entitled “Emerging Diagnostic Tests for COVID-19 Could Be on the 2 Horizon” provides a comprehensive review of current diagnostic tools for the detection of SARS-CoV-2.
Recently, a considerable number of works have reviewed the last advances in diagnostic methods for the monitoring of COVID-19 disease. The utilization of rapid point of care devices platforms is highly desirable to complement traditional diagnostic methods such as PCR and ELISA. This review presents the pros and cons of various analytical techniques to support the need for reliable diagnostic platforms for SARS-CoV-2 determination. However, the manuscript lacks clarity in its present form. Several sections require careful reorganization, and the comparative analysis of the different diagnostic approaches would help readers to understand the current state of art techniques. For these reasons, it is recommended the revision of the present version before publication.
I would also like to comment on the following issues that the authors should revise:
- The sentence in line 70 should be revised since the information contained is not completely accurate: “Transmission from contaminated surfaces has been identified as one of major source of infection in hospital set ups and in other affected areas”.
The structural and functional properties of the SARS-CoV-2 spike protein and the invasion mode of SARS-CoV-2 into host cells of the SARS-CoV-2 spike have been insufficiently described.
- The classification of diagnostic strategies for virus detection could be further clarified. For example, current diagnostic methods rely on the detection of the viral genome through nucleic acid amplification tests; direct detection of the intact virus via viral antigens, and indirect detection of antibodies or serological testing. However, rapid antigen detection is not mentioned in the text. Besides, although several biosensor strategies involved serological testing, they are considered in a separate section as other diagnostic techniques. Likewise, recent works on this subject have not been conveniently referenced.
- The discussion of biosensing applications could be improved by including updated references. Subsection 4.2. Spectroscopic biosensors include two works based on plasmonic related strategies, which were described in a previous section. Furthermore, the interferometry-based biosensor has not been referenced in the text.
- The information related to the fundamentals and analytical characteristics (sensitivity, selectivity) of the different techniques should be presented in Tables.
- Appropriate figures should illustrate the operating principles and configuration of the techniques presented.
- The works included are not sufficiently updated, several recent publications are not considered.
Reviewer 3 Report
The authors present a manuscript reviewing the employment and development of SARS-CoV-2 detection assays and sensors. The manuscript has grammar mistakes. It needs an English revision. The title “Emerging Diagnostic Tests for COVID-19 Could Be on the Horizon” seem to suggest that diagnostic tests are yet to come, whereas, the exact opposite is true, since there are several methods available already. This topic has been approached in quite an extension throughout literature in the recent two years; Despite, the obvious importance of assessing and evaluating the methods employed for detecting SARS-CoV-2 the proposed manuscript does not seem to bring much novelty or contribute in a significant matter that would warrant yet another review of this kind. Even on mdpi database, there are already over 10 review articles with similar proposals and topics in the 2020 and 2021. There is no methodology section.
It was recommended that the authors presented their review process with: Database searched (e.g. Web of Science, Science direct), terms used in the search, data range (year range for the search), number of articles retrieved, number of articles filtered (How many articles have been removed from the first withdrawn) and filter (The reason for the removal of the first withdrawn articles) and so forth. Reviews for MDPI journals, must follow the PRISMA guidelines.